# The evolution of critical thermal limits of life on Earth

Joanne M. Bennett [1,2,3,4✉], Jennifer Sunday [5], Piero Calosi[6], Fabricio Villalobos [7,8], Brezo Martínez[9], Rafael Molina-Venegas[10], Miguel B. Araújo [11,12], Adam C. Algar [13], Susana Clusella-Trullas[14], Bradford A. Hawkins[15], Sally A. Keith [16,17], Ingolf Kühn [1,4,18], Carsten Rahbek[17,18,19,20], Laura Rodríguez [9], Alexander Singer[1,2], Ignacio Morales-Castilla [10,21] & Miguel Ángel Olalla-Tárraga[9,21]

Understanding how species' thermal limits have evolved across the tree of life is central to predicting species' responses to climate change. Here, using experimentally-derived estimates of thermal tolerance limits for over 2000 terrestrial and aquatic species, we show that most of the variation in thermal tolerance can be attributed to a combination of adaptation to current climatic extremes, and the existence of evolutionary 'attractors' that reflect either boundaries or optima in thermal tolerance limits. Our results also reveal deep-time climate legacies in ectotherms, whereby orders that originated in cold paleoclimates have presently lower cold tolerance limits than those with warm thermal ancestry. Conversely, heat tolerance appears unrelated to climate ancestry. Cold tolerance has evolved more quickly than heat tolerance in endotherms and ectotherms. If the past tempo of evolution for upper thermal limits continues, adaptive responses in thermal limits will have limited potential to rescue the large majority of species given the unprecedented rate of contemporary climate change.

[1] German Centre for Integrative Biodiversity Research (iDiv) Halle-Jena-Leipzig, Leipzig, Germany. [2] Leipzig University, Ritterstraße 26, 04109 Leipzig, Germany. [3] Centre for Applied Water Science, Institute for Applied Ecology, Faculty of Science and Technology, University of Canberra, Canberra, Australia. [4] Institute of Biology, Martin Luther University Halle-Wittenberg, Halle (Saale), Germany. [5] Department of Biology, McGill University, Montreal, Canada. [6] Département de Biologie, Chimie et Géographie, Université du Québec à Rimouski, Rimouski, QC, Canada. [7] Departamento de Ecologia, Instituto de Ciências Biológicas, Universidade Federal de Goiás, Goiânia, Goiás, Brazil. [8] Red de Biología Evolutiva, Instituto de Ecología, A.C., Veracruz, México. [9] Department of Biology and Geology, Physics & Inorganic Chemistry, Universidad Rey Juan Carlos, Móstoles, Spain. [10] GloCEE - Global Change Ecology and Evolution Group, Department of Life Sciences, Universidad de Alcalá, Alcalá de Henares, Spain. [11] Department of Biogeography and Global Change, National Museum of Natural Sciences, CSIC, Madrid, Spain. [12] 'Rui Nabeiro' Biodiversity Chair, MED Institute, University of Évora, Évora, Portugal. [13] School of Geography, University of Nottingham, Nottingham, UK. [14] Centre for Invasion Biology, Department of Botany and Zoology, Stellenbosch University, Stellenbosch, South Africa. [15] Department of Ecology and Evolutionary Biology, University of California-Irvine, Irvine, USA. [16] Lancaster Environment Centre, Lancaster University, Lancaster, UK. [17] Center for Macroecology, Evolution and Climate, GLOBE Institute, University of Copenhagen, Universitetsparken 15, Copenhagen Ø 2100, Denmark. [18] Department Community Ecology, Helmholtz Centre for Environmental Research—UFZ, Halle, Germany. [19] Department of Life Sciences, Imperial College London, Ascot, Ascot SL5 7PY, UK. [20] Danish Institute for Advanced Study, University of Southern Denmark, Odense 5230 Odense M, Denmark. [21] These authors jointly supervised this work: Ignacio Morales-Castilla, Miguel Ángel Olalla-Tárraga. ✉email: Joanne.Bennett@canberra.edu.au

Understanding the geographic distribution of life on Earth is a core ecological research goal[1]. Across aquatic and terrestrial realms, connections between species' geographic range boundaries and their physiological thermal limits can help project the consequences of climate change on biodiversity[2,3]. For example, the tendency of species to retain their ancestral climatic affinities through evolutionary time is thought to constrain their ability to occupy climatic niches that differ from those occupied by their ancestors[4–6], limiting their potential to adapt to rapid warming.

The ability of organisms to tolerate cold temperatures is highly variable across species[7], clades[8] and geographic locations[9], while tolerance to heat is strikingly invariant across latitudes[10,11], elevation[12] and phylogeny[13]. This pattern is counter-intuitive when considering thermal fitness/performance curves, which are generally left-skewed: i.e. body temperature increases at higher temperatures have a much greater effect on fitness than the equivalent temperature decreases at lower temperatures. At the global scale, what causes upper thermal limits to be less variable across the entire tree of life than lower thermal limits[10,14] remains untested but could be elucidated by three distinctive but not mutually exclusive mechanisms[13].

First, 'deep-time climate legacies' would be detectable in thermal limits if species had a tendency to retain their ancestral climatic affinities under niche conservatism[4,6,15] and given the Earth's climate history. On a palaeoclimatic scale, Earth has been predominantly warm with intermittent glaciations[16,17], thus most clades may have originated during warm periods and may have had limited opportunities to evolve beyond the warm environmental conditions in which they arose[18]. If a 'deep-time climate legacy' is primarily responsible for the differential evolution of upper and lower tolerance limits, we expect both cold and heat thermal limits to be lower in cold-originating clades compared to warm-originating clades. By contrast, upper and lower thermal limits are expected to be higher in warm-originating clades.

Second, 'physiological boundaries' could limit physiological adaptation beyond certain temperatures. Lower thermal limits have been documented to exhibit more variation than upper thermal limits in both animals and plants[13], with a few exceptions emerging when considering intraspecific variation[19]. The inability of organisms to counter the destabilizing effects of high temperatures on membranes and proteins could constrain variation of heat tolerance beyond certain limits[13,20]. If so, this suggests the existence of a physiological boundary for heat (not necessarily for cold) tolerance. If as other physiological traits, such a boundary was evolutionary constrained, then it should be detectable using phylogenetic analyses that evaluate the tempo and mode of evolution of thermal limits. Specifically, if a boundary exists we would expect thermal physiological limits across clades to accumulate through time consistent with an Orstein–Uhlenbeck (OU) model of evolution which indicates a stabilising selection on species thermal tolerance traits towards a fitness optimum[21]. Conversely, if neither a physiological boundary nor fitness optimum exist, the evolution of thermal limits may better fit a random model of evolution such as Brownian Motion, where a trait evolves in a random walk process, or white noise, where the trait value varies independently around the global mean[22]. While slower rates of evolution are expected for heat tolerance, because it is less variable than cold tolerance[13], we still lack a phylogenetically informed multi-taxon test for this hypothesis. If a physiological boundary or optimum is the primary driver of invariance in upper thermal limits, we would expect the mode of evolution to show aggregation around an upper limit value, consistent with an evolutionary constraint for tolerance to heat. A lack of such aggregation would be expected for lower thermal limits.

Third, 'adaptation to current climatic extremes' is expected to exert selective pressure on thermal limits. However, we expect current climatic extremes to exert greater selection on lower compared to upper thermal limits for two reasons. On the one hand, maximum environmental temperatures tend to be less variable across contemporary biogeographic gradients (i.e. latitude) compared to minimum temperatures[9,23,24]. On the other hand, behavioural buffering is more likely to reduce selective pressure on heat tolerance relative to cold tolerance[15], because while organisms are able to use behaviour to evade heat stress, there tends to be fewer opportunities to behaviourally evade cold stress[5,15]. If 'adaptation to current climatic extremes' is the main determinant of species' thermal tolerance limits, we would expect a close match between experienced and tolerated temperature extremes[9] and similar rates of evolution in upper and lower thermal limits, with no global optimum or boundary.

Here, using the largest existing dataset for thermal limits[25], we conduct a series of comprehensive analyses to disentangle the relative role of the above-mentioned mechanisms that may have contributed to shape the global variation in thermal physiological limits. We stress again that these mechanisms are not mutually exclusive, especially given that species distributions can shift through time to remain within a thermal niche, allowing evolutionary constraints and current climates to determine distributions (e.g.[26]). We investigate these mechanisms using proxy variables. To investigate species' geographic and temporal distributions in relation to changes in climate at evolutionary timescales, we determine (a) the 'thermal ancestry' of every species in our dataset, based on the palaeoclimatic conditions predominant on Earth at the time when its order originated[17]: as a proxy variable for the 'deep-time climate legacies' hypothesis. We consider ancestry at the order level in an attempt to disentangle the effects of temperature at clade origin from a time for speciation effect[27]. We also determine (b) the evolutionary age of each species' order (to assess evolutionary constraints associated to the 'physiological boundaries' hypothesis); and (c) current thermal regimes experienced across species' ranges (minimum and maximum environmental temperatures) as proxies of 'adaptation to current climatic extremes' (see Supplementary Note 3 for details). Indeed, our results show that 'adaptation to current climatic extremes' is the strongest determinant of species thermal tolerance limits, while there simultaneously remains a signal consistent with a 'deep-time climatic legacy' in the cold tolerance of ectotherms, and a signal of evolutionary constraints on 'physiological boundaries' for heat tolerance across all groups investigated.

## Results

**Biogeography of thermal tolerance limits.** Thermal physiological limits in our dataset are homogeneously represented across latitudes (ranging from 70°S to 70°N), while thermal ancestry is heterogeneously distributed across latitudes (Fig. 1a). The distribution of data across aquatic and terrestrial realms reflects the distribution of life on earth where ~80% of macroscopic species are terrestrial[28] (Fig. 1b). Because of the differences in data availability across realms, with fewer samples of aquatic taxa, we pool aquatic and terrestrial data of ectotherms and endotherms (but not plants due to broader phylogenetic disparity) for subsequent analyses. Thermal physiological limits include lethal and critical thermal limits of plants (i.e. photosynthetic plants and macroalgae) and ectotherms, and edges of thermal neutral zones (TNZ) in endotherms (for more details see the "Methods" section and Supplementary Note 1). We compare evolutionary patterns in upper and lower thermal physiological limits for lethal and critical thermal limits of plants and ectotherms, and edges of TNZ

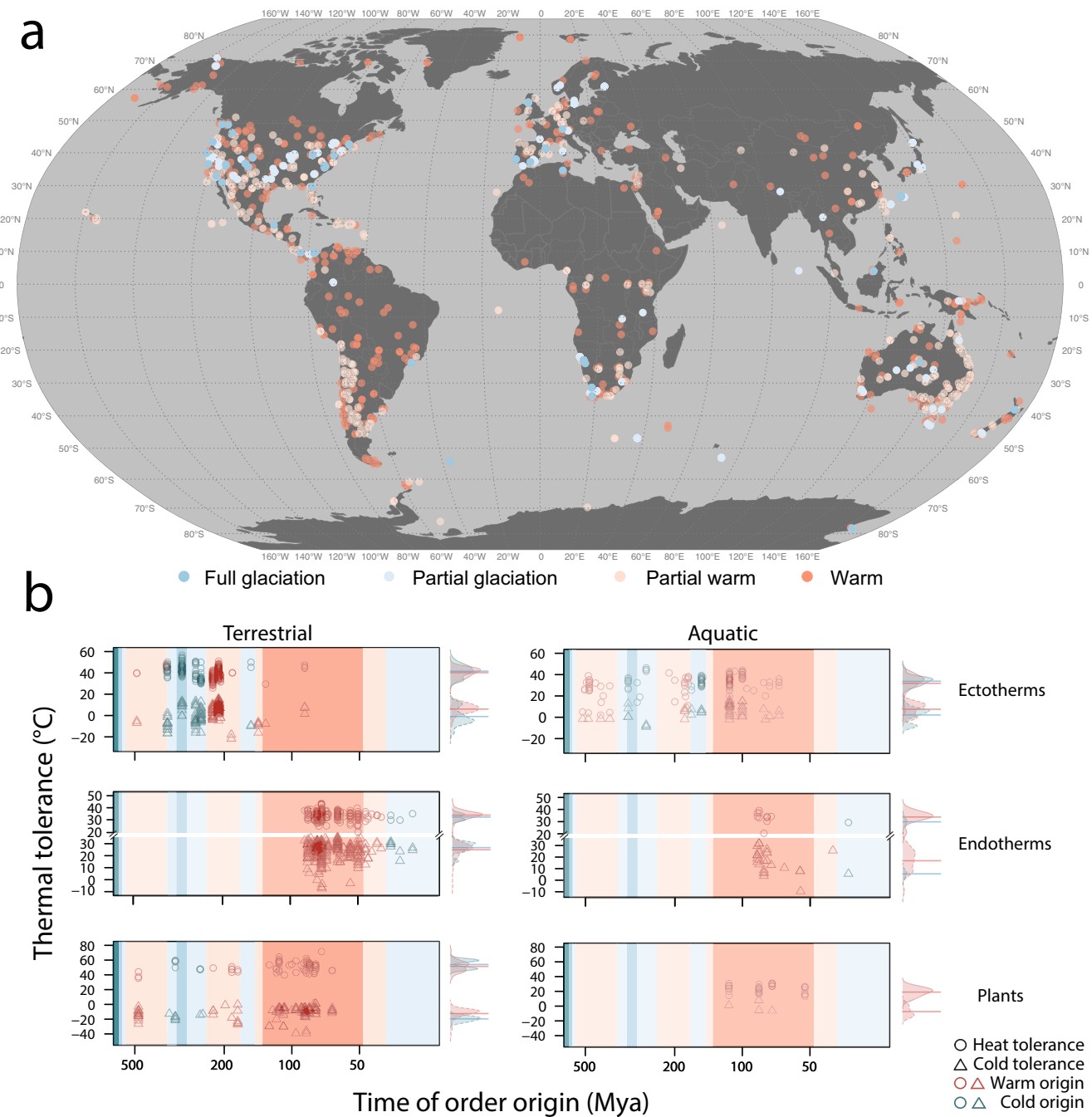

**Fig. 1 A map illustrating the geographic location at which experimental specimens were collected and plots of the relationship between order age and thermal tolerance limits.** Map points (**a**) and the plot area shading (**b**) are coloured according to the prevailing paleoclimate at order origin (see Supplementary Fig. 2) full glaciation (blue), partial glaciation (light blue), partial warm (light orange) and warm (orange). **b** The raw relationships between order age in million years (mya) and lower (triangles) and upper (circles) thermal tolerance limits for ectotherms, endotherms and plants. Points (**b**) are coloured red for species in warm origin orders (partial warm and warm palaeoclimate categories as shown in the map) and blue for species in cold origin orders (full and partial glaciation palaeoclimate categories as shown in the map). For endotherms only (**b**), the axis is broken so that upper and lower thermal limits can be clearly delineated. Density distributions of upper and lower thermal tolerance limits are shown to the right of each panel **b**, aggregated by time of origin, following the same colour scheme as above, lines correspond to medians. Source data are provided as a Source Data file.

separately as they each interact differently with species' physiology, behaviour, and environmental conditions (see Supplementary Note 1). We found that variation within lower and upper thermal limits increases with clade age, more clearly in ectotherms and endotherms than in plants (Fig. 1b). However, this variation may be due to sampling as *n* decreases in both ectotherms and endotherms towards most recent times. Most ancestors of the species in our dataset (~80%) originated under warm climatic regimes. Species with ancestors that originated under glaciation times are mostly sampled across temperate latitudes (~80%) with the majority of those in the Northern hemisphere (~80%) (Fig. 1).

**Paleoclimate origin and thermal tolerance.** Ectotherm and terrestrial plant species belonging to orders that originated during

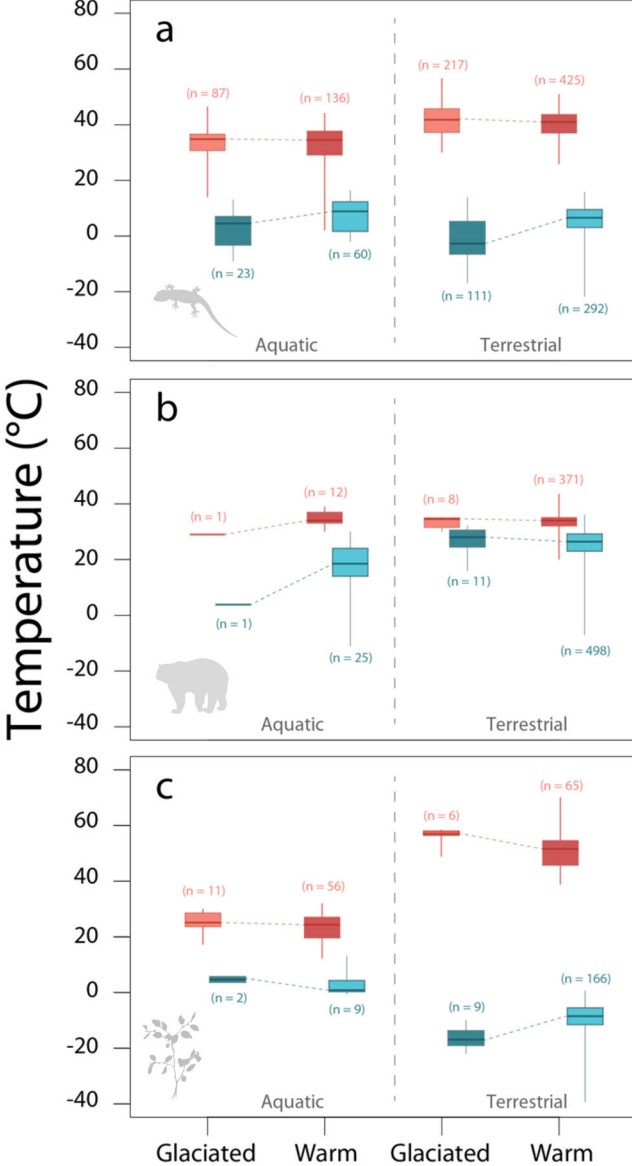

**Fig. 2 Test of the effect of deep-time climate legacies.** The boxplots compare the distributions of upper (red) and lower (blue) thermal tolerance of species belonging to orders of terrestrial and aquatic (**a**) ectotherms, (**b**) endotherms and (**c**) plants (photosynthetic plants and macroalgae). Dark colours reflect the palaeoclimatic conditions of order origination expected to show either lower values in lower thermal limits (darker blue for species belonging to orders originated under glaciated palaeoclimates—data from partial glaciated and glaciated paleoclimate categories combined) or higher values in upper thermal limits (darker red for species belonging to orders originated under warm non-glaciated palaeoclimates—warm and partial warm paleoclimate categories combined). For details on data collection see Supplementary Note 1. Boxes are bounded within the first and third quartiles, medians represented by thick horizontal lines within each box and, whiskers extending to the minimum and maximum values that do not exceed 1.5 times the interquartile range from the median (provided by default in R function 'boxplot'). Source data are provided as a Source Data file.

glacial periods tend to show lower cold tolerance compared with those originating from periods when Earth was predominantly warm (Fig. 2). This pattern of variation in cold tolerance is consistent with the 'deep-time climate legacies' hypothesis, showing that species whose ancestors originated under glacial

palaeoclimates tend to tolerate colder temperatures than species of warm thermal ancestry. On the contrary, a warmer or colder origin of ancestry does not seem to influence mean or variance in heat tolerance (Fig. 2 and Supplementary Table 1; see also Fig. 1b and Supplementary Fig. 1) indicating that factors other than conservatism of climatic conditions must shape the variation in these thermal traits. However, a colder origin of ancestry appears not to apply, broadly speaking, to endotherms and aquatic plants (Fig. 1b, Supplementary Table 1 and Supplementary Fig. 1). For details and results at other taxonomic levels see Supplementary Table 2.

**Evolutionary age and thermal tolerance limits.** We found that lower thermal limits consistently evolved faster than upper limits across taxa (Table 1, also see Supplementary Note 4). This supports the existence of physiological constraints that affect the tempo of evolution of upper compared to lower thermal limits. This pattern of asymmetric evolutionary rates appears to have configured over relatively recent evolutionary time as it emerges at the family level or below (see Supplementary Table 2). The differences are markedly larger for endotherms than for ectotherms or plants (Fig. 3a–c; see also Supplementary Fig. 2).

Variation in species' thermal physiological limits across clades accumulated through time in a more consistent manner with an OU model of evolution than with the alternative Brownian Motion or White Noise models (Table 1; see also Supplementary Note 2 and Supplementary Fig. 2). Fits for the $\alpha$ parameter suggest a moderate to strong directional selection ($-\log \alpha \ll 0$) towards "attractor" values—i.e. or phenotypic values with non-random higher frequencies, suggesting they are selected for (Table 1). Greater support for models with this mode of evolution (i.e. greater log-likelihood values for OU compared to alternatives) was consistent across taxonomic groups and levels (Supplementary Tables 1 and 2) showing selection towards an attractor, in both upper and lower thermal limits, regardless of whether only one (Supplementary Table 4) or several thermal tolerance metrics were combined within an analysis (Table 1). The results show that selection towards an 'attractor' is stronger on upper thermal limits than lower thermal limits in endotherms and terrestrial plants (lower $-\log \alpha$ in Table 1), but a comprehensive comparison is limited by data availability (see Supplementary Notes 2.3 and 5). Support for OU patterns is commonly interpreted as evidence for either stabilising selection or phylogenetic niche conservatism[21]. An alternative interpretation of our results would be directional selection—i.e. towards attractor phenotypes—having acted together with a physiological barrier, constraining the evolution of thermal tolerances beyond a threshold. Our results clearly show strong phylogenetic structure in tolerance to both heat and cold; however, our data are a limited sample of the full tree of life and therefore, we recommend caution when trying to infer evolutionary processes from our results. For further details on phylogenetic results and assumptions, see Supplementary Note 2.

**Relative hypothesis support.** Using random forests to compare all three hypotheses invoked to explain thermal limits we found current minimum and maximum environmental temperatures experienced by species[29], to play a strong role in determining thermal tolerance variation, consistent with the importance of 'adaptation to current climatic extremes' (Fig. 4, Supplementary Tables 5 and 6). Both lower and upper thermal limits increased with current environmental temperatures for all taxonomic groups, although the proportion of variance explained in endotherms was much lower than for ectotherms and plants (see Supplementary Fig. 5). However, for ectotherms and endotherms

**Table 1 Tempo and mode of evolution of upper and lower thermal tolerances of ectothermic species, endothermic species, terrestrial plants and plants and algae.**

| Taxa | Metric | n | TEMPO ($\sigma^2$) | MODE ($-\log \alpha$) | LnLik OU | LnLik BM | LnLik WN |
|---|---|---|---|---|---|---|---|
| Ectotherms | Upper | 547 | 0.784 ± 0.225 | −0.983 ± 0.113 | −1542.68 ± 42.711 | −1577.036 ± 57.495 | −1787.285 ± 0 |
| | Lower | 335 | 1.224 ± 0.385 | −1.126 ± 0.112 | −1007.873 ± 30.696 | −1042.516 ± 41.797 | −1098.014 ± 0 |
| Endotherms | Upper | 314 | 0.593 ± 0.153 | −1.262 ± 0.029 | −817.683 ± 10.032 | −885.844 ± 23.664 | −830.211 ± 0 |
| | Lower | 495 | 2.067 ± 0.182 | −1.029 ± 0.05 | −1598.035 ± 9.283 | −1666.438 ± 19.03 | −1655.963 ± 0 |
| Plants | Upper | 32 | 0.675 ± 0.171 | −0.835 ± 0.084 | −103.014 ± 0.486 | −106.764 ± 1.841 | −104.089 ± 0 |
| | Lower | 71 | 1.183 ± 0.353 | −0.494 ± 0.237 | −235.901 ± 4.283 | −241.399 ± 7.372 | −250.602 ± 0 |
| Plants & Algae | Upper | 81 | 1.366 ± 0.366 | −0.56 ± 0.131 | −260.011 ± 6.033 | −265.575 ± 7.898 | −328.995 ± 0 |
| | Lower | 77 | 1.3 ± 0.355 | −0.659 ± 0.209 | −261.85 ± 3.793 | −269.043 ± 7.242 | −276.223 ± 0 |

The tempo measures the rate of thermal tolerance evolution (in °C Mya$^{-1}$), and the mode informs of the likelihood within which a given model of evolution fits the data (for details, see Supplementary Note 2). Note that n (the number of species within each grouping) varies according to the number of taxa for which there are records in GlobTherm and which are included in the phylogenetic hypothesis used for analyses[59]. The phylogenetic hypothesis are Ornstein Uhlenbeck (OU), Brownian Motion (BM) or White Noise (WN) models. Source data are provided as a Source Data file.
LnLik log-likelihood of model.

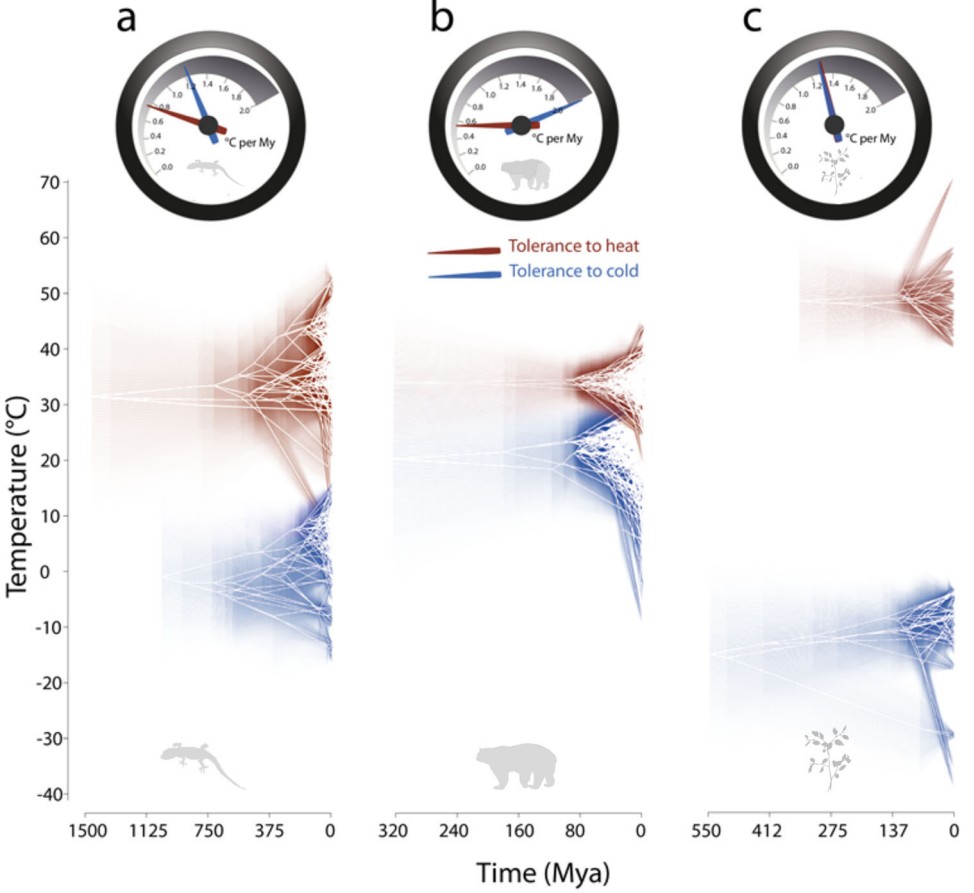

**Fig. 3 Tempo and mode of evolution of thermal tolerance limits.** Tempo and mode of evolution of upper (red) and lower (blue) thermal tolerance limits of **a** ectotherms, **b** endotherms and **c** plants (photosynthetic plants and macroalgae). The top velocimeters illustrate the rate of evolution as measured by $\sigma^2$. Estimates of $\sigma^2$ are computed as the average between the results for the smoothed and unsmoothed phylogenetic trees in ref. [18]. Sample sizes and details on the uncertainty around the estimates are supplied in Supplementary Tables 2–4 and see Supplementary Note 2. The bottom traitgrams together with the uncertainty about ancestral character states shown by increasing transparency illustrate the phenotypic change along evolutionary time.

the age of the origin clade was similarly or more important than current climate conditions in explaining variation in thermal tolerance limits (Fig. 4, Supplementary Tables 5 and 6) although the direction of the tolerance–age relationship varied across taxa and limits (see Supplementary Fig. 6a–d for details). To test the 'deep-time climate legacies' hypothesis we included the predominant palaeoclimate category at which orders originated.

Palaeoclimate ranked the lowest across all factors tested and only emerged as a significant variable for ectotherms: for which, its importance reached 5.9–5.3% for upper and lower limits, respectively (Fig. 4a, Supplementary Fig. 5 and Supplementary Tables 5 and 6). While the relative importance of palaeoclimate is rather low even in ectotherms the direction of its associations with thermal tolerance coincide with that predicted: 'cold-origin'

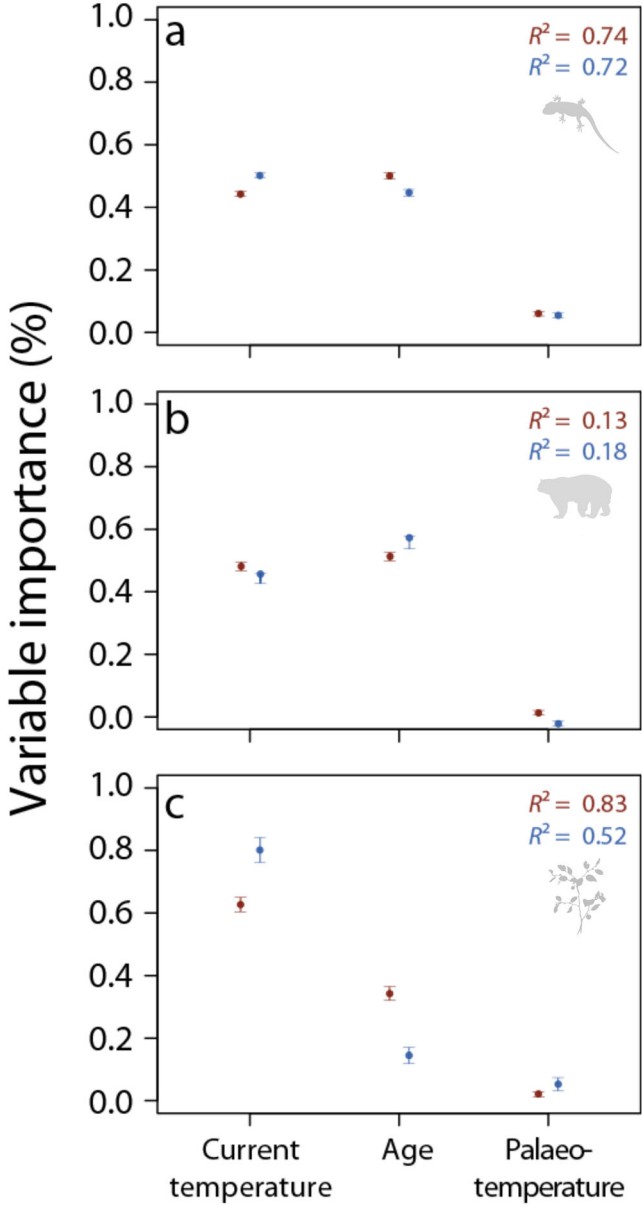

**Fig. 4 The importance of experienced contemporary climate, clade evolutionary age and palaeoclimatic origin in predicting thermal limits.** Variable importance in random forest models fitting the relationships between upper (red) and lower (blue) thermal limits and predictors including palaeoclimatic origin (palaeo-temperature), experienced contemporary climate (current temperature) and clade evolutionary age (age) for **a** ectotherms, **b** endotherms, **c** plants (combing data from aquatic and terrestrial realms). Average model accuracy ($R^2$) is reported for each model subset. Error bars represent 95% confidence intervals. For source data and sample sizes see Supplementary Table 5 and Supplementary Note 3.

species better withstand colder temperatures than 'warm-origin' species, and 'warm-origin' species withstand slightly warmer temperatures than 'cold-origin' ones (see Supplementary Fig. 5a, b, and Supplementary Note 3 for further details of analysis).

Together, our results show that present-day environmental temperatures, but also that physiological constraints in the tempo of evolution (in ectotherms and endotherms) and, to a much lesser extent, the climate at clade origin (for cold limits in ectotherms) affect cold and heat tolerances.

## Discussion

Our results offer insights into the evolution of thermal tolerance. We reject the hypothesis of Brownian Motion—i.e. accumulation of phenotypic variation linearly proportional to the evolutionary time elapsed—as the underlying mode of evolution of both upper and lower thermal limits (Table 1), which is often assumed in comparative studies. Instead, our results indicate that upper thermal limits evolve towards an attractor value consistent with an OU model of evolution (Table 1). This result is consistent with the hypothesis of selection towards an upper physiological boundary that is not readily crossed, or an optimum beyond which fitness declines[20]. Indeed, although experimental evidence is sparse, experimental findings for drosophilid flies show limited evolutionary capacities to evolve heat resistances >39 °C[14]. The 'attraction' of upper thermal limits in metazoans around a relatively narrow range possibly reflects absolute constraints due to oxygen limitation and/or the existence of a shared thermal sensitivity of macromolecular structures[7,30]. Furthermore, the tempo of evolution suggests that both upper and lower limits are phylogenetically constrained, with upper limits evolving more slowly though the tree of life. Our findings are consistent with the existence of a limited scope for further adaptation to increasing heat resistance in a rapidly warming planet[31]. This, coupled with the narrower thermal safety margins that tropical ectothermic species display[32], provide a cause for concern that thermal tolerance evolution will most likely not rescue populations from climate change-driven extinction[33,34].

Our results also offer insight into macroevolutionary patterns of clades across Earth. Because the majority of clades evolved during warm periods, the species-poor and cooler higher latitudes appear to offer opportunities for speciation and evolution of thermal tolerance through adaptive radiation[13,35]. Indeed, increases in thermal breadth over time have been driven by changes in lower rather than upper thermal limits, which have evolved more slowly (Fig. 3, Supplementary Table 5), and remained more constant than lower thermal limits through time (see Supplementary Fig. 8; see also Fig. 2). These findings are consistent with the hypothesis that there are greater opportunities for speciation and evolution of thermal tolerances in cold environments and perhaps with reverse speciation gradients i.e. lower speciation in the tropics[36].

Cold tolerance in endotherms has evolved remarkably quickly compared to cold tolerance in ectotherms and plants, however this possibly reflects the different determinants of TNZ of endotherms compared to thermal limits of ectotherms and plants. For instance, biophysical models have shown that TNZ of endotherms extend to colder temperatures with increasing body size and the thickness of fur insulation[37], but no relationship of size on critical thermal limits (CTmin) exists in lizards[38]. Evolutionary changes in cold tolerance of ectotherms might involve a series of interacting biochemical changes[39], which may take much longer to evolve than changes in fur length, feather depth or body size in endotherms.

Our study provides a broad-scale formal test of the long-standing hypotheses that species thermal limits are conserved[40], and that physiological constraints limit the evolution of heat tolerance[20] while considering them alongside other putative mechanisms: i.e. current climate extremes. We find that these three mechanisms all effect species thermal tolerance limits, but their relative importance varies. Specifically, species' thermal tolerance limits appear to be strongly linked to current climate, but there is also evidence supporting the existence of 'physiological boundaries' to the evolution of upper temperature tolerance across all groups, and a small (but consistent) effect of temperature of clade origin in cold tolerance for ectotherms. Ultimately, our finding that thermal limits are constrained by evolution, and

conserved through time across broad taxonomic groups, can inform and improve predictions about how species would redistribute under warmer or colder climates. Although thermal tolerance has arisen as a central trait to assess species vulnerability amidst the ongoing biodiversity crisis, additional traits, such as thermoregulatory regimes and behaviour, add layers of refinement that will further improve our ability to more accurately project species' distributions under future climate change[3,5].

## Methods

**Data**. Experimentally derived thermal tolerance limit data were obtained from GlobTherm[25], which assembles published measurements of upper and lower thermal tolerance limits, including both lethal and critical thermal metrics for plants and ectotherms and the edges of TNZ in endotherms. Lethal thermal limits mark the temperature when mortality occurs. Critical thermal limits record the temperature at which a key an ecological function is lost, such as locomotion, or as in the case of endotherms the ability maintain basal metabolism (i.e. TNZ). GlobTherm is a database of comparable thermal tolerance metrics with broad taxonomic coverage across terrestrial and marine realms. The data collection protocol considered comparability between studies and ameliorated known quality issues associated with the determination of lethal temperature for algae and the boundary of the TNZ for mammals and birds. Specifically, GlobTherm only contains lethal limit data from studies where all temperature treatments were indicated, and only contains TNZ data from studies showing evidence that the upper or lower boundary of the TNZ was reached. Here, we only used data for algae if the interval between measurements was ≤2 °C, to overcome the difficulties associated with determining where death occurred within the interval. This gave us 2038 species for analysis. Classification of realms (marine, intertidal, freshwater and terrestrial) followed the IUCN Red List of Threatened Species version 3[41], World Register of Marine Species WoRMS[42] and AlgaeBase[43]. Animal species were broadly defined as ectothermic (lineages other than mammals and birds) or endothermic (mammals and birds). Geographic coordinates reflect latitude and longitude of the location at which the experimental organisms were collected. Based on the coordinates supplied in GlobTherm we extracted air temperature at 2.5 min resolution for terrestrial taxa (WorldClim version 2.1 climate data for 1970–2000)[44] and sea surface temperature at 5 min resolution for marine taxa (bio-ORACLE v2.0 encompassing the 2000–2014)[45,46]. Estimates of clade age were extracted from the evolutionary time tree of life[18], the largest, most comprehensive calibrated tree that exists to date[47]. Our grouping of clades into palaeoclimate categories followed ref. [17]: (1) full glaciation, (2) partial glaciation, (3) partial warm, and (4) warm, to reduce errors associated with matching clade ages to climate estimates in deep geological time. The dating of the palaeoclimate categories is based on a broad consensus of the major deep-time climate trends of the Earth's history (Fig. 1)[16,17]. For more on the caveat associated with deep time climate trends see Supplementary Note 5. To better disentangle the effects of temperature at clade origin from a time for speciation effect—i.e. the fact that clades that have existed for longer would be more diverse simply due to having had longer times to diversify[27], results presented in the main text are focused at the order level (for details and results at other taxonomic levels see Supplementary Table 2). The order level was chosen because high taxonomic ranks (i.e. order level) have been shown to align with phylogenetic temporal banding (e.g. the absolute dates of evolutionary origin) and provide homogeneous units of comparison at the taxonomic level for phenotypic divergence, as it is in our case[48]. Further, taxonomic classifications are more robust at high taxonomic levels (i.e. order and above) compared to lower taxonomic levels[49]. Thus, species were assigned to the palaeoclimatic level corresponding to the taxonomic order to which they belong. For further details on data collection see Supplementary Note 1.

To increase taxonomic coverage and sample size in ectotherms and plants (photosynthetic plants and macroalgae), here we present results using both lethal and critical thermal limits, although limits of TNZ were exclusively analysed in endotherms. Patterns observed were robust when analyses were conducted on data subsets for single measurements of thermal tolerance (i.e. lethal or critical limits), and variation in experimental design was taken into account: i.e. ramping rate and pre-treatment acclimation temperature, which is available for only a subset of the data. See Supplementary Fig. 7 and Supplementary Table 6, and Supplementary Note 5.

**Phylogenetic analyses**. We tested the tempo and mode of evolution of upper and lower thermal tolerances of ectotherms, endotherms and plants, following common practice in evolutionary biology[50]. We computed the Brownian rate parameter $\sigma^2$ as an indicator of the rate of evolution[51]. $\sigma^2$ provides an estimate of the accumulation through time of phenotypic variation[51], for both tolerance to heat and to cold. Following ref. [50], we tested and compared the likelihoods of three of the most common models of character evolution: Brownian Motion, Ornstein Uhlenbeck (OU), and a White Noise model representing a *null* model where phenotypic variation evolves at random. For OU models, we compared the parameter $\alpha$, which measures the strength with which trait evolution tends towards an attractor value. We validated results obtained from the tree of life[18], with results from different phylogenetic hypotheses for subsets of the taxa (e.g. plants[52], amphibians[53],

reptiles[54]), and with results summarised across 100 trees sampled from the Bayesian posterior distribution of phylogenetic trees when available (e.g. birds[55] and mammals[56]). More specifics on the assumptions and expectations of phylogenetic analyses can be accessed in Supplementary Note 2 and caveats in Supplementary Note 5.

**Statistical analyses**. To explore the relationships between upper or lower thermal tolerances and palaeoclimatic origin, biogeographic location, and evolutionary age, we fitted random forest models, a machine-learning method that corrects data overfitting[57] and allows non-linear relationships. We fitted the models with 500 decision trees and used node purity values to inform the importance of each predictor. The method iteratively samples bootstrapped subsets of data that are subsequently employed to fit decision trees. Results are then averaged, informing of the relative importance of each predictor and their errors, which are decreased with respect of those in individual decision trees. Because random forest outputs consist of as many decision trees as specified (i.e. 500), we display one fitted classification tree for each model and indicate within it the relative importance of the variables from the random forest (see Fig. 4). A model was fitted for each combination of upper or lower thermal tolerance and each group of ectotherms, endotherms and plants (photosynthetic plants and macroalgae). We report random forest results in the main text for simplicity, but alternative modelling procedures (e.g. Bayesian Hierarchical models) confirmed these results qualitatively. For further details on statistical analyses and results see Supplementary Note 3. All analyses were conducted in R version 4.0.3[58].

**Reporting summary**. Further information on research design is available in the Nature Research Reporting Summary linked to this article.

## Data availability

The dataset of species thermal physiological limits analysed during the current study is available in the Dryad repository, doi:10.5061/dryad.1cv08. For more details on the dataset, see Bennett et al. (2018)[25]. Classification of realms followed the IUCN Red List of Threatened Species version 32 (available at: http://www.iucnredlist.org)[41], World Register of Marine Species WoRMS (available at: http://www.marinespecies.org)[42] and AlgaeBase (available at: http://www.algaebase.org)[43]. Estimates of clade age were extracted from the evolutionary time tree of life available at http://www.timetree.org[59]. Additional phylogenies used for validation included: plants (Dryad repository, doi:10.5061/dryad.63q27)[60], amphibians (Dryad repository, doi:10.5061/dryad.vd0m7)[61], squamate (Dryad repository, doi: 10.5061/dryad.82h0m)[62], birds (Bird tree depository, http://birdtree.org)[55] and mammals from the supplementary data file provided by Faurby and Svenning (2015)[56]. Source data are provided with this paper.

## Code availability

The associated analysis codes are archived on github (https://github.com/MoralesCastilla/ThermalEvolution/tree/v1.0, https://doi.org/10.5281/zenodo.4311705)[63]

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

## Acknowledgements

This is a joint effort of the sWEEP working group supported by sDiv, the Synthesis Centre of the German Centre for Integrative Biodiversity Research (iDiv) Halle-Jena-Leipzig (DFG FZT 118, 02548816). We are indebted to the work of the hundreds of researchers who have published the results of their experiments on thermal limits for the wide array of taxa encompassed here. J.M.B was supported by a sDiv postdoctoral fellowship. I.M.-C. and M.A.O.-T. acknowledge funding by the Spanish Ministry of Science and Innovation (Grant PID2019-109711RJ-I00 to I.M.-C., Grant CGL2017-86926-P to M.Á. Rodríguez and Grant CGL2017-89820-P to M.A.O.-T.). J.M.S. acknowledges funding from the Natural Sciences and Engineering Council of Canada. P.C. acknowledges funding from the Natural Sciences and Engineering Council of Canada Discovery Grant Program (RGPIN-2020-05627) and he is member of the FRQ-NT network Québec-Océan.

## Author contributions
The majority of authors discussed and formulated the overall idea of the work at a joint workshop. J.M.B. collected all the data, led the data analyses and writing of the manuscript. J.S. assisted in the analyses and writing of the manuscript. P.C. assisted in the writing of the manuscript. F.V. assisted in the analyses and writing of the manuscript. B.M. assisted in writing of the manuscript. R.M-V. assisted in analyses and writing in the manuscript. M.B.A. assisted in the writing of the manuscript. A.C.A. contributed to drafts of the manuscript. S.C.T. contributed to drafts of the manuscript. B.A.H. contributed to drafts of the manuscript. S.K. contributed to drafts of the manuscript. I.K. contributed to drafts of the manuscript. C.R. contributed to drafts of the manuscript. L.R. contributed to drafts of the manuscript. A.S. contributed to drafts of the manuscript. I.M-C. was a principal investigator on the project, led analyses in the manuscript and the writing of the manuscript. M.A.O-T. was a principal investigator on the project, led analyses in the manuscript and the writing of the manuscript.

## Funding

## Competing interests
The authors declare no competing interests.
