## [Peer Review File · Nature Communications]

Reviewers' Comments:

Reviewer #1:

Remarks to the Author:

This paper uses a recently compiled--and large-- data set on heat and cold tolerances, adds an explicitly historical perspective. It finds, for example, that contemporary species belonging to clades that originated during cold periods tend to have lower cold tolerances than those belonging to clades that originated in warm periods. It also finds that heat tolerance appears to evolve more slowly than does cold tolerances: the paper interprets this to suggest that a "hard physiological boundary" exists, which will imply an evolutionary barrier to adaptation to climate warming.

For me the most interesting aspect was the idea of "deep-time climate legacies." A finding of evolutionary stasis of heat tolerance and of optimal temperatures has been noted and discussed many times over the decades, but the analyses here are novel (e.g., Bogert '49 stated — without any evidence — that thermal preferences of lizards trace as far back as the Miocene!), but I don't recall any explicit attempt to tie of tolerances to deep-time climates. That is very interesting, and raises lots of questions for future 'ponderings.' Overall the paper develops a number of novel analyses of classic and contemporary themes in thermal ecology.

Given that some of these issues have been around, it would seem appropriate that a few classical papers (beyond just Janzen '67) should be cited. As is, almost all of the non-methodological citations are restricted to the last decade or so, giving the misleading impression that these are contemporary issues only. However, the idea of a hard physiological boundary -- as an example -- was central to W. Hamilton's (1973) "maxithermy" hypothesis.

"plants inclusive of photosynthetic algae (hereafter plants)". Somewhat unclear writing. In any case, the Supplement clarifies that you included only algal groups. Calling them "plants" is very misleading and implies that your results are general for plants, which cannot be the case. Why not just refer to them as algae?
Also, the idea of 'niche conservatism' in thermal physiology (thought not the term) traces at least to Bogert CM (1949). Thermoregulation in reptiles, a factor in evolution. *Evolution* 3, 195-211 and has been discussed many times. For thermal biology, see papers on the Bogert Effect by Huey et al. and Muñoz et al.

For morphological parallels, see Wake DB, Roth G, Wake MH (1983). On the problem of stasis in organismal evolution. *Journal of Theoretical Biology* 101, 211-24.

A fourth is that maximum temperatures are much similar across latitude (at least at latitudes with many species) than are minimum temperatures. Buckley LB, Huey RB (2016). Temperature extremes: geographic patterns, recent changes, and implications for organismal vulnerabilities. *Global Change Biology* 22, 3843-58.

Also, ectotherms exposed to cold stress are usually in retreats and unable to use behavior to evade that stress, and so selection on cold tolerance can be strong. In contrast, ectotherms exposed to heat stress can often use behavior (movement) to evade that stress. Therefore, behavioral buffering (sensu Bogert, Muñoz) can lead to relative stasis of heat tolerance.

Interesting point, but why should that be so, given lineages seemingly responded physiologically to intermittent glaciations? If they could evolve differences then, why not now?

76ff This is an old idea and has been discussed many times ever since Hamilton raised the concept of 'maxithermy', and there are many (old) empirical examples in frogs, lizards, insects.

Hamilton, WJ, III (1973). *Life's color code*. McGraw Hill, New York, N.Y.

Hamilton, W.J., III and C. G. Coetzee. 1969. Thermoregulatory behaviour of the vegetarian lizard *Angolosaurrrs skoogi* on the vegetationless northern Namib Desert dunes. *Sri. Pap Nutrib Desert Res. Srn. NO.47:95- 10*

Heinrich, B (1977). Why have some animals evolved to regulate a high body temperature? *American Naturalist* 111, 623-40. (was critical of Hamilton)

a few empirical examples:

Brattstrom, B. H. 1968. Thermal acclimation in anuran amphibians as a function of latitude and altitude. *Comp. Biochem. Physiol.* 24:93-111.

Snyder GK, Weathers WW (1975). Temperature adaptations in amphibians. *American Naturalist* 109, 93-101.

van Berkum FH (1988). Latitudinal patterns of the thermal sensitivity of sprint speed in lizards. *American Naturalist* 132, 327-43.

also, Payne & Smith

As an aside, some cyanobacteria reportedly photosynthesize up to 75°C (Castenholz 1969), suggesting some groups have escaped the 'hard boundary.'

see Hoffmann AA (2010). Physiological climatic limits in *Drosophila*: patterns and implications. *Journal of Experimental Biology* 213, 870-80.
need citation here.

84-7 The issue of lower relative evolutionary rates of high-temperature tolerance limits is interesting. Because fitness/performance curves are skewed, a 1° increase in temperature (at high temperature) has a much greater effect of fitness than does a 1°C decrease (at low temperature). Therefore, I would think (a priori) that this would lead to stronger selection on heat tolerance — however reasonable that seems to me, it is not supported by the comparative data (relatively limited evolution of heat tolerance) that is presented here and in earlier papers!

Fig. 1 & 89 Interesting idea and finding! What happens if you use a more 'recent' taxonomic division (family)? You address your logic to some extent (line 242), and on 118 you state that the results are robust to taxonomic level — but I missed any supporting evidence.

The lower panels in Fig. 1 are much wider than high, which 'squashes' existing variation in thermal limits and thus biases the image.

Are modern representatives of clades that originated in cold (or warm) climates more likely to be found (today) in cold (or warm) climates? That seems to be a direct assumption here. If I understand line 267, you should be able to see whether palaeoclimatic origin predicts contemporary biogeographic location.

Line 217 refers to lethal and critical thermal metrics, but these are not relevant to endotherms. In the supplement, you mention that you used "edge of the thermal neutral zone." I think that must be noted in the main text, as that metric is very different from that used here for ectotherms. If you include endotherms, you should be explicit (in the main text) that you are using the thermal neutral zone for endotherms. There's precedent of course, but CTmax, CTmin are very different from Tu,crit and Tl, crit.

Perhaps more importantly, the limits of the thermal neutral zone of endotherms are related strongly to body size, shape, and fur — none of which is controlled here (unless I missed something) The importance of size and fur has been discussed at least since Scholander et al. 1950, and has been modeled by Porter & Kearney a decade ago.

Porter WP, Kearney M (2009). Size, shape and the thermal niche of endotherms. *Proceedings of the*

National Academy of Sciences, USA 106, 19666-72Å.

Size does not have a comparable effect on heat or cold tolerance of ectotherms (as far as I recall), but certainly it does for endotherms. I'm ok with your analyses for ectotherms and algae, but not those for endotherms.

Incidentally, the upper limit of the thermal neutral zone of endotherms is constrained, but for issues pertaining to heat balance (see Porter & Kearney) . That constraint does not apply to heat tolerance of ectotherms. So you are grouping apples and oranges by including endotherms.

In Fig 1 and in Fig S2, the lower limits for "plants" are generally well below 0°C. I'm not an algal physiologist, but I'm rather skeptical that algae photosynthesize at such levels.

Edit this sentence

132 "maxithermy"

150 Buckley LB, Huey RB (2016). How extreme temperatures impact organisms and the evolution of their thermal tolerance. *Integrative and Comparative Biology* 56, 98-109.
Hoffmann (above).

184 "maxithermy"

197 See some of the empirical papers cited above (Brattstrom, Snyder & Weathers, etc.)

204 they increase with body temperature, not necessarily ambient

240 "to align with phylogenetic temporal banding" — unclear^[1]_[SEP]
Do your ideas still hold if you re-scale temperatures onto a thermodynamic scale? See Payne & Smith.

Supporting information

9 Please explain how limiting temperatures were determined for photosynthesis — when photosynthesis went to 0? Chronic vs. acute exposures, which will probably generate different limits.

Delete "Hedges et al" — use only the citation number
Hedges' et al. but Zenne et al.'s — inconsistent. you would write using the phylogeny of Zenne et al.
or of Hedges et al.

"sensitive amounts"?

Overall, I find this to be an interesting and significant paper, despite a few misgivings.

Ray Huey

Reviewer #2:

Remarks to the Author:

Review of Bennett et al "The Evolution of Critical Thermal Limits of Life on Earth"

The current manuscript presents an impressive attempt to explore the evolution of thermal tolerance under a phylogenetically-informed context by leveraging a huge dataset on thermal limits across much of the tree of life. I think that the dataset is impressive as are many of the analyses, and I think that

some of the results are compelling and interesting. That said, I have a number of concerns about the data and analyses that I think need to be addressed. I first describe my overall/large concerns and then provide some line comments.

Major points

1. I am concerned that the measures of critical thermal limits compiled and used as dependent variables in analyses are not sufficiently commensurate, and thus that it is not possible to draw strong conclusions from the analyses. For example, righting-response CT_{max} is commonly measured in lizards, while thermal limits are frequently inferred from growth-rate data in insects or LT₅₀ is used. Moreover, many thermal limit measures are well documented to be affected by rate of heating/cooling (e.g. Rezende et al 2014 *Funct Ecol* 28:799-809), which does not appear to be accounted for. Thus, any methodological biases by taxon could greatly bias your conclusions. The measures used are sufficiently different that I am not convinced that they can be pooled to form a single dependent variable for analysis. I think that these differences need to be expressly accounted for in the primary analyses, or preliminary analyses demonstrating that biases do not actually exist are needed. Moreover, although numerous caveats are provided in the supplemental section 4, this one is not, even though it could have marked impacts on the conclusions.

2. I am concerned that the OU model is not appropriate for modeling the evolution of high-temperature limits. The OU model assumes that traits are continuous and normally distributed around the "attractor" value (e.g. Butler and King *Am Nat* 164: 683-695 and Blomberg 2017 *BioRxiv* <https://doi.org/10.1101/067363>). This implies that trait values above and below the attractor occur, but are selected against which pulls the mean toward the attractor (i.e. the rubber band). Especially for maximal thermal limits, I think that this assumption is strongly broken, which could affect your ability to draw inferences from the model. The lack of extremely heat-tolerant species should result in a strongly left-skewed character distribution that would be poorly modeled by an OU process. Moreover, your hypothesis (2) of a hard physiological barrier, actively suggests that trait values must be left-skewed as they run into the barrier. Unfortunately, I am not aware of a good solution for this problem except for simulating evolution of traits responding to directional selection with a physiological "wall" and analyzing them with an OU model to discover how the model behaves. Alternatively, you could use an alternative modeling strategy such as that outlined by Blomberg 2017 *BioRxiv* <https://doi.org/10.1101/067363>.

3. For examining the climate legacies hypothesis, I think that more than glaciation status of the planet needs to be taken into consideration. Even during glaciations, some locals were warm and during interglacials some locals were cold. I think that this spatial variation needs to be accounted for in order to test climate legacies (e.g. paleo-biogeography rather than just current biogeography). I think that only using time is too coarse for testing this hypothesis.

4. As written, each of the three hypotheses proposed to explain the evolution of thermal limits (LNS 72-84) seem overly simplistic. Moreover, they are pitched as alternative hypotheses, yet all are apparently supported by the data (LNS 48-51). I think that the paper would benefit from some rethinking of this framework with more nuanced hypotheses that better represent current knowledge in the field. For example, when considering hypothesis (1), even though taxa may have evolved in relatively "thermally stable" warm conditions, there is still sizeable variation present in the thermal environment resulting in organisms facing challenging temperatures. Current/recent environments present frequent selection where tolerance of higher temperatures would be beneficial. Thus hypothesis 1, as written, seems to require organisms never, or at least rarely, be subject to thermal challenge, which is not the case. For hypothesis (2), I agree that the literature indicates that upper thermal limits are less variable than lower thermal limits, but there is still variation which argues against a simple physiological barrier as described. Within animals alone, vertebrates seem to max out at ~46C (dessert iguana) although insects can commonly survive into the 50's C and the Pompeii worm can survive to ~80C. So "hard boundaries" as described already appears to be falsified. This is

not to say that there would not be important evolutionary constraints that could be taxon specific or driven by phylogenetic inertia. However, this is a more nuanced hypothesis than what is presented. For hypothesis (3), the argument that “adaptation to rare extreme events” would explain the difference in variance between high and low temperature tolerance seems problematic because it ignores the difference in high and low temperature safety margins. Thermal performance curves are generally left-skewed resulting in smaller safety margins at high temperature. As a result, extreme warm events of less absolute magnitude can have larger effects on fitness than cold events of larger absolute magnitude. Also, the most important determinant of cold tolerance is frequently freeze tolerance, which is not currently considered. All this to say that there are numerous a priori reasons to think that simple differences in the absolute magnitude of high and low-temperature thermal variation would not adequately explain the evolution of thermal tolerance.

Line Comments

LNS 78-80: This clause seems out of place/unrelated to the prior clause.

LNS 95-96: This sentence is hard to follow as written, and seems to be missing a verb.

LNS 120-125: Although the observation that there is more variation in cold tolerance than heat tolerance and more taxa originated in “warm” time periods is consistent with the deep-time climate legacies hypothesis, I do not find it compelling. We knew a priori that cold temperature tolerance is more variable than high-temperature tolerance, and there were only two possible paleoclimate environments considered (i.e. warm or cold). Thus, there was a 50:50 chance of observing this. Also, it doesn’t account for the known plasticity of cold thermal tolerance.

LNS 142-144: I think that this statement is too strong given that the only alternative models considered were Brownian motion and white noise. All that you can say is that the data were more consistent with an OU model. I’m not sure you can distinguish the difference between an optimum “attractor” and a wall however. I’m also not sure how well OU does with boundary conditions, given that this breaks one of the assumptions of the model (as described above).

LNS 157-160: I don’t think that observed min and max temps are really a proxy for “extreme events” which are usually defined as being abnormal events (e.g. more than 2 sd from the mean). Looking at the max and min tell you nothing about the distribution of thermal conditions. Also, I cannot find details for the environmental max and min values used. Are these air temperatures or operative temperatures? Air temperatures can do a fairly good job of modelling min but are often quite different from the possible max experienced by organisms. Some additional description of these data would be useful.

LNS 207-208: I disagree that the analyses presented provide a mechanistic understanding of thermal evolution. We neither gain insight into the mechanisms driving selection nor the physiological processes most directly under selection/evolving. Rather, these analyses and their inferences seem correlative/statistical. For example, current and paleo-climates were correlated with thermal tolerance and variance. This is interesting, but I do not think it can accurately be called mechanistic.

Reviewer #3:

Remarks to the Author:

This study provides a large-scale phylogenetic analysis of ‘thermal limits’ across the tree of life, and provides the same sort of message of previous reviews that heat tolerance somehow has reached a physiological boundary and may not be able to evolve in response to climate change. In this sense, the main take-home message is not entirely new. Nonetheless, one might still argue that analyses

would be of interest to the general community if results were reliable, but unfortunately this is not the case. Analyses are oversimplistic and ignore a multitude of confounding factors that should be taken into consideration from the general quality of the data to the classification employed to reconstruct the paleoclimate associated with the origin of each evolutionary lineage.

Without going any further, there are at least two fundamental problems with the study in terms of the quality of the data: (1) thermal limit estimates vary substantially with measurement protocol and there are no attempts to control for confounding effects and (2) estimates of so-called 'thermal limits' between ectotherms and endotherms are by no means comparable.

With regards to the first point, there is extensive literature on the problems associated with the estimation of 'thermal limits' and their accuracy that spans more than a decade. To make a long story short, Lutterschmidt and Hutchison (1997) pointed out in the first review on the subject that critical endpoints were highly sensitive to differences in measurement protocol, and this problem was brought back to light following the empirical paper by Terblanche et al. (2007). Subsequently, Rezende et al. (2014) showed that, other confounding effects aside, previous comparative studies have ignored the impact of exposure time and that this variable accounts for nearly 64% of the variation across organisms/studies. This limitation remains in the current submission, thermal limit estimates are not directly comparable and consequently any sort of phylogenetic reconstruction that neglects the fact that we might be comparing apples and oranges will be biased and unreliable. Considering the diversity of data that is available at Globtherm, the bare minimum that is required here is to account for potential confounding effects such as (i) exposure time associated with the endpoint estimation, (ii) experimental protocol (ramping versus static) and (iii) developmental stage.

For the second point, the authors pool estimates of thermal limits in ectotherms (i.e., endpoints where animals lose coordination, are knocked down or die; or what some researchers call 'ecological death') with the limits of the thermoneutral zone (TNZ) in endotherms, which are simply not comparable. The limits of the TNZ correspond to the temperatures in which endotherms must elevate their metabolism above basal levels to thermoregulate at rest, and this is far from a thermal estimate of 'ecological death' for an endotherm. Accordingly, only a handful of endothermic species can apparently cope with temperatures below 0 °C in Fig 3b, while in reality many birds and mammals regularly encounter lower temperatures in their ranges of distribution. Yes, I understand that some studies have used TNZ limits as a standardized proxy to estimate thermal limits in endotherms, which is problematic but at least comparisons were performance across endothermic species. Here, the authors argue that these estimations are comparable to CTmax or other estimates of thermal limits in ectotherms, and this is simply wrong. TNZ limits have been adopted as a standard measure of thermal tolerance for endotherms for practical purposes because there are real estimations of CTmax or CTmin for birds and mammals, which is not to say that these estimates are in any way comparable to the endpoints where locomotion is disrupted or lethal temperatures. This explains, for instance, the counterintuitive result in Fig 3 that ectotherms can tolerate colder temperatures than endotherms when one of the main arguments to explain the evolution and diversity of endothermic lineages was to colonize cold environments and distribution patterns would suggest the opposite trend (see Grady et al 2019).

Ultimately, this study involves the use of some sophisticated phylogenetic statistical methods to reconstruct the evolution of thermal tolerance based on a global dataset that was already available, without any consideration regarding the quality of the data, what different estimates actually estimate, etc. Major sources of uncertainty underlying the distribution of thermal tolerance estimates have been completely dismissed, and therefore I am afraid analyses and results are simply not reliable.

References

Grady et al. (2019) Metabolic asymmetry and the global diversity of marine predators. *Science* 25:363

Lutterschmidt, W.I. & Hutchison, V.H. (1997) *Canadian Journal of Zoology*, 75, 1553–1560.

Rezende et al (2014) Tolerance landscapes in thermal ecology. *Funct Ecol* 28: 799-809.

Terblanche, J.S., Deere, J.A., Clusella-Trullas, S., Janion, C. & Chown, S.L. (2007) *Proceedings of the Royal Society B: Biological Sciences*, 274, 2935–2942.

Reviewers' comments:

Reviewer #1 (Remarks to the Author):

This paper uses a recently compiled--and large-- data set on heat and cold tolerances, adds an explicitly historical perspective. It finds, for example, that contemporary species belonging to clades that originated during cold periods tend to have lower cold tolerances than those belonging to clades that originated in warm periods. It also finds that heat tolerance appears to evolve more slowly than does cold tolerances: the paper interprets this to suggest that a “hard physiological boundary” exists, which will imply an evolutionary barrier to adaptation to climate warming.

For me the most interesting aspect was the idea of “deep-time climate legacies.” A finding of evolutionary stasis of heat tolerance and of optimal temperatures has been noted and discussed many times over the decades, but the analyses here are novel (e.g., Bogert '49 stated — without any evidence — that thermal preferences of lizards trace as far back as the Miocene!), but I don't recall any explicit attempt to tie of tolerances to deep-time climates. That is very interesting, and raises lots of questions for future ‘ponderings.’ Overall the paper develops a number of novel analyses of classic and contemporary themes in thermal ecology.

Given that some of these issues have been around, it would seem appropriate that a few classical papers (beyond just Janzen '67) should be cited. As is, almost all of the non-methodological citations are restricted to the last decade or so, giving the misleading impression that these are contemporary issues only. However, the idea of a hard physiological boundary -- as an example -- was central to W. Hamilton's (1973) “maxithermy” hypothesis.

RE: We thank the Reviewer for their positive comments on our work, and in particular for recognising the novelty of our analytical approach with regard to testing for the presence of “deep-time climate legacies”. Following the Reviewer suggestion, we have conducted a new search of old literature, and have now cited more relevant references (incl. Bogert et al. 1949 and Hamilton 1973). We also introduce in the main text the concept of maxithermy (lines 80-84): “(2) strong physiological constraints could limit physiological adaptation beyond certain temperatures, for example, the inability of organisms to counter the destabilizing effects of high temperatures on membranes and proteins may lead to an optimum operational temperature i.e. Hamilton's ‘maxithermy’ hypothesis¹⁷.”

“plants inclusive of photosynthetic algae (hereafter plants)”. Somewhat unclear writing. In any case, the Supplement clarifies that you included only algal groups. Calling them “plants” is very misleading and implies that your results are general for plants, which cannot be the case. Why not just refer to them as algae?

RE: Thank you for pointing out this potentially unclear aspect of our paper. Indeed, our analyses included both plants (Phylum Streptophyta) and algae (Phylum Chlorophyta, Rhodophyta, Phaeophycacea). To compare results in these taxa at the broad scale (i.e. to ectotherms and endotherms), we grouped photosynthetic taxa as “plants” for several analyses and presentations. Our text now states (lines 52-54):

“Ectothermic animals and photosynthetic organisms (plants and algae, hereafter ‘plants’) from clades originated in cold paleoclimates have lower cold tolerance limits than those, which originated in warm periods”

In the supplementary file 1 lines 21-23. “We grouped photosynthetic organisms (plants and macroalgae) together in most analyses and plots (hereafter grouped as “plants”).”

Also, the idea of ‘niche conservatism’ in thermal physiology (thought not the term) traces at least to Bogert CM (1949). Thermoregulation in reptiles, a factor in evolution. *Evolution* 3, 195-211 and has been discussed many times. For thermal biology, see papers on the Bogert Effect by Huey et al. and Muñoz et al.

For morphological parallels, see Wake DB, Roth G, Wake MH (1983). On the problem of stasis in organismal evolution. *Journal of Theoretical Biology* 101, 211-24.

RE: Great point. We have looped into this literature with two new reference additions (although we’d like to add them all, we are limited by reference number restrictions): Huey, R. B. et al. Predicting organismal vulnerability to climate warming: roles of behaviour, physiology and adaptation. *Philos. Trans. R. Soc. B Biol. Sci.* 367, 1665–1679 (2012).

Wake, D. B., Roth, G. & Wake, M. H. On the problem of stasis in organismal evolution. *J. Theor. Biol.* 101, 211–224 (1983).

A fourth is that maximum temperatures are much similar across latitude (at least at latitudes with many species) than are minimum temperatures. Buckley LB, Huey RB (2016). Temperature extremes: geographic patterns, recent changes, and implications for organismal vulnerabilities. *Global Change Biology* 22, 3843-58. Also, ectotherms exposed to cold stress are usually in retreats and unable to use behavior to evade that stress, and so selection on cold tolerance can be strong. In contrast, ectotherms exposed to heat stress can often use behavior (movement) to evade that stress. Therefore, behavioral buffering (sensu Bogert, Muñoz) can lead to relative stasis of heat tolerance.

RE: We agree and have now incorporated latitude and behaviour into *Hypothesis 3* with this relevant citation.

Revision lines 87-93: “(3) Adaptation to survive rare and extreme climatic events, given that maximum environmental temperatures tend to be less variable across contemporary biogeographic gradients (i.e. latitude) compared to minimum temperatures^{9,18,19}. Further, behavioural buffering is more likely to reduce selective pressure on heat tolerance relative to cold tolerance²⁰, because while organisms are able to use behaviour to evade heat stress, there tend to be fewer opportunities to evade cold stress^{5,20}.”

Interesting point, but why should that be so, given lineages seemingly responded physiologically to intermittent glaciations? If they could evolve differences then, why not now?

RE: although species may have survived cold climates, the time spent in these climates was shorter, such that heat tolerance may not have been lost, and novel cold tolerance mechanisms may not have had time to be gained.

76ff This is an old idea and has been discussed many times ever since Hamilton raised the concept of 'maxithermy', and there are many (old) empirical examples in frogs, lizards, insects.

RE lines 80-84: We have now cited Hamilton's maxithermy hypothesis. "(2) Strong physiological constraints could limit physiological adaptation beyond certain temperatures, for example, the inability of organisms to counter the destabilizing effects of high temperatures on membranes and proteins may lead to an optimum operational temperature i.e. Hamilton's 'maxithermy' hypothesis¹⁷."

Hamilton, WJ, III (1973). Life's color code. McGraw Hill, New York, N.Y.

Hamilton, W.J., III and C. G. Coetzee. 1969. Thermoregulatory behaviour of the vegetarian lizard *Angolosaurris skoogi* on the vegetationless northern Namib Desert dunes. Sri. Pap Nutrib Desert Res. Srn. NO.47:95- 10

Heinrich, B (1977). Why have some animals evolved to regulate a high body temperature? American Naturalist 111, 623-40. (was critical of Hamilton)

a few empirical examples:

Brattstrom, B. H. 1968. Thermal acclimation in anuran amphibians as a function of latitude and altitude. Comp.

Biochem. Physiol. 24:93-111.

Snyder GK, Weathers WW (1975). Temperature adaptations in amphibians. American Naturalist 109, 93-101.

van Berkum FH (1988). Latitudinal patterns of the thermal sensitivity of sprint speed in lizards. American Naturalist 132, 327-43.

also, Payne & Smith

RE: We thank the Reviewer for pointing to these useful references. We have now added to the text the references by Hamilton, Brattstrom and Payne & Smith, but due to restrictions in the number of references that we are allowed we could not add all.

As an aside, some cyanobacteria reportedly photosynthesize up to 75°C (Castenholz 1969), suggesting some groups have escaped the 'hard boundary.'

RE: We agree with the reviewer that unicellular organisms (not covered in our present analysis) have 'escaped' the hard molecular and functional boundaries that evolution appears to have set for multicellular organisms, but boundaries (although higher) have also been suggested for unicellular organisms: see Storch et al. 2014 GCB. Additionally, some multicellular organisms have (exceptionally) evolved to live under extreme thermal conditions (incl. a restricted number of species among the phyla Pogonophora, Annelida, Mollusca and Arthropoda that have adapted to thermal regimes and extremes associated with hydrothermal vents, also as pointed by Reviewer #2 for the annelid *Alvinella pompejana*), but what we want to know is if these examples are "exceptions" to a strong evolutionary tendency towards an optimum (OU model), or if it is the norm through Brownian motion. We have changed the

language though out the manuscript to clarify this point and now refer to this hypothesis as “Strong physiological constraints” rather than a “hard boundary”.

see Hoffmann AA (2010). Physiological climatic limits in *Drosophila*: patterns and implications. *Journal of Experimental Biology* 213, 870-80.

RE: The suggested citation was added

need citation here.

RE: We have added the citation: Buckley LB, Huey RB (2016). Temperature extremes: geographic patterns, recent changes, and implications for organismal vulnerabilities. *Global Change Biology* 22, 3843-58.

84-7 The issue of lower relative evolutionary rates of high-temperature tolerance limits is interesting. Because fitness/performance curves are skewed, a 1° increase in temperature (at high temperature) has a much greater effect of fitness than does a 1°C decrease (at low temperature). Therefore, I would think (a priori) that this would lead to stronger selection on heat tolerance — however reasonable that seems to me, it is not supported by the comparative data (relatively limited evolution of heat tolerance) that is presented here and in earlier papers!

RE: We thank the reviewer for these helpful insights and have added this insightful point in text (lines 71-76) “This pattern is counter-intuitive when considering thermal fitness/performance curves, which are generally left-skewed so that temperature increases at higher temperatures have a much greater effect of fitness than the equivalent temperature decrease at lower temperatures.”

Fig. 1 & 89 Interesting idea and finding! What happens if you use a more ‘recent’ taxonomic division (family)? You address your logic to some extent (line 242), and on 118 you state that the results are robust to taxonomic level — but I missed any supporting evidence.

RE: We have now included the results that show that taxonomic level did not affect the tempo or mode of evolution in the supplementary information, please see Table S2.

The lower panels in Fig. 1 are much wider than high, which 'squashes' existing variation in thermal limits and thus biases the image.

RE: We have reformed the Fig. 1 as per the reviewer suggestion.

Are modern representatives of clades that originated in cold (or warm) climates more likely to be found (today) in cold (or warm) climates? That seems to be a direct assumption here. If I understand line 267, you should be able to see whether palaeoclimatic origin predicts contemporary biogeographic location.

RE: Great point. We have updated the manuscript to include the suggested analysis.

Revision lines 214-217 supplementary file: “Species collected at higher latitudes were more often originated at a time the Earth was glaciated than those collected closer to

the equator (for lower thermal limits $F(1,1) = 18.0$, $p < 0.001$ and for upper thermal limits $F(1,1) = 35.8$, $p < 0.001$).”

Line 217 refers to lethal and critical thermal metrics, but these are not relevant to endotherms. In the supplement, you mention that you used “edge of the thermal neutral zone.” I think that must be noted in the main text, as that metric is very different from that used here for ectotherms. If you include endotherms, you should be explicit (in the main text) that you are using the thermal neutral zone for endotherms. There’s precedent of course, but CTmax, CTmin are very different from Tu,crit and TI, crit. Perhaps more importantly, the limits of the thermal neutral zone of endotherms are related strongly to body size, shape, and fur — none of which is controlled here (unless I missed something) The importance of size and fur has been discussed at least since Scholander et al. 1950, and has been modeled by Porter & Kearney a decade ago.

Porter WP, Kearney M (2009). Size, shape and the thermal niche of endotherms.

Proceedings of the National Academy of Sciences, USA 106, 19666-72.

Size does not have a comparable effect on heat or cold tolerance of ectotherms (as far as I recall), but certainly it does for endotherms. I’m ok with your analyses for ectotherms and algae, but not those for endotherms.

Incidentally, the upper limit of the thermal neutral zone of endotherms is constrained, but for issues pertaining to heat balance (see Porter & Kearney). That constraint does not apply to heat tolerance of ectotherms. So you are grouping apples and oranges by including endotherms.

RE: We thank the reviewer for these insights into the factors determining thermal limits in endotherms. We agree that the thermal neutral zone of endotherms are related strongly to body size and shape, and our inability to account for these variables is key to our interpretation of the endotherm analyses. Most importantly, we run analyses separately on endotherms, ectotherm and plants so that the different experimental methods employed to measure thermal tolerance in these taxonomic groups can be evaluated separately. Although we present results from apples and oranges together, we make strong attempts not to group them at any point in the paper, and have revised our text to improve clarity on this point.

To better articulate the different limit types used, we have revised the text (lines 232-233) which now reads as: “including both lethal and critical thermal metrics for ectotherms and the edge of the thermal neutral zone in endotherms”.

In Fig 1 and in Fig S2, the lower limits for “plants” are generally well below 0°C. I’m not an algal physiologist, but I’m rather skeptical that algae photosynthesize at such levels.

RE: Text revisions have been made to clarify that the analysis includes both plants and algae. The majority of thermal limit data for plants comes from lethal measures in land plants, i.e. the limit for survival not the limit for photosynthesis, and such measures commonly go well below zero.

RE: Main text lines 52-54: “Ectothermic animals and photosynthetic organisms (plants and algae, hereafter ‘plants’) from clades originated in cold paleoclimates have lower cold tolerance limits than those, that originated in warm periods.”

In the supplementary file 1 lines 21-23. “We grouped photosynthetic organisms (plants and macroalgae) together in most analyses and plots (hereafter grouped as “plants”).

95 Edit this sentence

RE: We have revised this sentence for clarity, revision lines 101-104: “To investigate species’ geographic and temporal distributions in relation to changes in climate at evolutionary timescales, we documented (a) the ‘thermal ancestry’ of every species in our dataset, based on the palaeoclimatic conditions predominant on Earth at the time when its ancestor originated (see methods)¹⁵”

“maxithermy”

RE: As explained above, we have revised the manuscript in lines 83-85. We have also revised lines 138-142 to incorporate this reference, which now reads: “Additionally, if a strong physiological boundary is the primary driver of invariance in upper thermal limits, we would expect the mode of evolution to show aggregation around a given upper limit value, suggesting that an optimum (i.e. maxithermy¹⁷) or a constraint on evolution actually exists for tolerance to heat.”

Buckley LB, Huey RB (2016). How extreme temperatures impact organisms and the evolution of their thermal tolerance. *Integrative and Comparative Biology* 56, 98-109. Hoffmann (above).

RE: We have now included the suggested reference line 87-89: “Adaptation to survive rare and extreme climatic events, given that maximum environmental temperatures tend to be less variable across contemporary biogeographic gradients (i.e. latitude) compared to minimum temperatures^{9,18,19}.”

184 “maxithermy”

RE: We have revised the manuscript in lines 81-84 and 138-142 to incorporate this theory.

197 See some of the empirical papers cited above (Brattstrom, Snyder & Weathers, etc.)

RE: We have revised the manuscript to include many of the suggested references including Brattstrom.

204 they increase with body temperature, not necessarily ambient

RE: Good point. We have revised the text to clarify this in lines 215-219: “This pattern could be explained by either Janzen’s climatic variability hypothesis², which predicts species will have broader thermal ranges in more seasonally variable environments, or by metabolic and biochemical reactive rates increasing exponentially with body temperature, limiting thermal breadth in warmer latitudes³³.”

240 “to align with phylogenetic temporal banding” — unclear

Do your ideas still hold if you re-scale temperatures onto a thermodynamic scale? See Payne & Smith.

RE: We agree with the reviewer’s suggestion it would be very interesting for future work to ask if thermal limits are responding to temperature or to thermodynamic rates, although this is beyond the scope of our aims. If rescaled to thermodynamic rates, it seems logical that rates of evolution in upper thermal limits would be found to be greater relative to the rates of evolution in cold limits within taxonomic groups. However, this would throw into question why evolution of thermal limits would act on thermodynamic rates rather than on temperature per se. This is a very interesting idea and worthy of exploration in a separate analysis. We have revised the text in lines 258-261 to improve clarity and explain phylogenetic temporal banding, which now reads: “High taxonomic ranks (i.e. order level) have shown to align with phylogenetic temporal banding (e.g. the absolute dates of evolutionary origin) providing homogeneous units of comparison for phenotypic divergence, as it is in our case³⁹”.

Supporting information

9 Please explain how limiting temperatures were determined for photosynthesis — when photosynthesis went to 0? Chronic vs. acute exposures, which will probably generate different limits.

RE: The vast majority of data for plants and algae comes from lethal measures, which are typically determined *via* static (chronic) exposures rather than dynamic (acute). Whilst we agree that chronic and acute measures will most likely generate different limits, Chronic measures were the most commonly available data for photosynthetic organisms. Critical measures relating to photosynthesis did not generally go beyond 0 but commonly do for lethal measures. The text has been updated to highlight that acute measures are rare for plants, and our analysis has now been modified so that we also present analysis that only includes the major limit type within each group (see supplementary file section 3. Statistical analyses).

RE lines 14-18 supplementary material: “Different experimental measures are often associated with different groups of organisms and in part, this is why we analysed endotherms (all measured using TNZ), plants and algae (almost exclusively lethal experiments), ectotherms (majority are critical limits) separately (for more information on the phylogenetic analysis see below).”

Delete “Hedges et al” — use only the citation number Hedges’ et al. but Zenne et al.’s — inconsistent. you would write using the phylogeny of Zenne et al. or of Hedges et al.

RE line 97 and 99 the suggested revision was made.

“sensitive amounts”?

RE lines 107-108 supporting information now reads “We did not reanalyse the tempo and mode of evolution of thermal limits for arthropods nor for algae because well resolved comprehensive published phylogenies were not available”.

Overall, I find this to be an interesting and significant paper, despite a few misgivings.

Ray Huey

Reviewer #2 (Remarks to the Author):

Review of Bennett et al “The Evolution of Critical Thermal Limits of Life on Earth”

The current manuscript presents an impressive attempt to explore the evolution of thermal tolerance under a phylogenetically-informed context by leveraging a huge dataset on thermal limits across much of the tree of life. I think that the dataset is impressive as are many of the analyses, and I think that some of the results are compelling and interesting. That said, I have a number of concerns about the data and analyses that I think need to be addressed. I first describe my overall/large concerns and then provide some line comments.

Major points

1. I am concerned that the measures of critical thermal limits compiled and used as dependent variables in analyses are not sufficiently commensurate, and thus that it is not possible to draw strong conclusions from the analyses. For example, righting-response CTmax is commonly measured in lizards, while thermal limits are frequently inferred from growth-rate data in insects or LT50 is used. Moreover, many thermal limit measures are well documented to be affected by rate of heating/cooling (e.g. Rezende et al 2014 *Funct Ecol* 28:799-809), which does not appear to be accounted for. Thus, any methodological biases by taxon could greatly bias your conclusions. The measures used are sufficiently different that I am not convinced that they can be pooled to form a single dependent variable for analysis. I think that these differences need to be expressly accounted for in the primary analyses, or preliminary analyses demonstrating that biases do not actually exist are needed. Moreover, although numerous caveats are provided in the supplemental section 4, this one is not, even though it could have marked impacts on the conclusions.

RE: We agree with the reviewer that different experimental measures are commonly associated with different groups of organisms. For this reason, we took steps to collate metrics that would be most comparable *within* groups, so that evolutionary rates *within* them would make sense. In plants and ectothermic animals, these were exclusively dynamic ramping endpoints of critical organismal function (CT), and lethal limits under static conditions (LT), and we did not combine these with other definitions of thermal limits e.g. those based growth rate endpoints, or others based on specific physiological functions (such as heart pumping or respiration rates). We have updated the supplementary text in lines 14-18 to clarify our reasoning and to make our methods clearer: “Different experimental measures are often associated with different groups of organism and in part, this is why we analysed endotherms (all measured using TNZ), plants and algae (almost exclusively lethal experiments),

ectotherms (majority critical limits) separately (for more information on the phylogenetic analysis see below)".

The reviewer highlighted that there could be an issue with mixing lethal and critical measures in our analysis. To determine if this was the case, we have now analysed these groups separately where appropriate and have included these new analyses in the supplementary material (Table S5-S6). We found that separating these groups did not change the interpretation of our results, therefore to maximise taxonomic coverage we have continued to discuss these groups together in text, while referring to the supplementary results.

We agree with the reviewer with regards to the other aspects of the experimental design (i.e. ramping rate and acclimation temperature) that are known to affect the experimental endpoint. We understand the reviewer's concern and realize that we should have been more explicit about the influence that different experimental treatments might have had on our results. This issue only affects estimates from ectotherms and plants, while endotherms use a standard experimental approach. With regards to ectotherms and plants we have now included in the supplementary material results from a Random Forest model that includes pretreatment temperature and ramping rate as covariates. We found that although ramping rate and pre-treatment temperature are important in determining thermal tolerance, they did not change our conclusions qualitatively. We have updated the text (see lines: 310-319 supplementary file) and added Table S7 showing these results. We also cite a recent paper by some of the authors of this ms. where analyses explicitly explore the effects of experimental treatments (Sunday et al. 2019) and similarly found that ramping rate and pretreatment do not have a major effect on global patterns in thermal tolerance. Revision lines 310-319: "The Globtherm dataset was designed to be a database of comparable thermal tolerance metrics. To achieve it's goal, preference was given to sources that used the most widely used methods and measurements and only included data from adults. However, differences in experimental design can affect the estimates of thermal end-point, this is particularly relevant to ectotherms when experiments use different ramping rates and pre-treatment temperatures. Data on ramping rate and pre-treatment temperature is available for a limited number of taxa in GlobTherm and including such treatments as covariates in our Random forests did not change our conclusions qualitatively (Table. S7). Our finding is consistent with previous studies which have shown the magnitude of the effect of ramping rate and pre-treatment temperature do not affect the interpretation of global patterns in thermal limits using the Globtherm dataset31."

2. I am concerned that the OU model is not appropriate for modeling the evolution of high-temperature limits. The OU model assumes that traits are continuous and normally distributed around the "attractor" value (e.g. Butler and King Am Nat 164: 683-695 and Blomberg 2017 BioRxiv <https://doi.org/10.1101/067363>). This implies that trait values above and below the attractor occur, but are selected against which pulls the mean toward the attractor (i.e. the rubber band). Especially for maximal thermal limits, I think that this assumption is strongly broken, which could affect your ability to draw inferences from the model. The lack of extremely heat-tolerant species should result in a strongly left-skewed character distribution that would be poorly modeled by an OU process. Moreover, your

hypothesis (2) of a hard physiological barrier, actively suggests that trait values must be left-skewed as they run into the barrier. Unfortunately, I am not aware of a good solution for this problem except for simulating evolution of traits responding to directional selection with a physiological “wall” and analyzing them with an OU model to discover how the model behaves. Alternatively, you could use an alternative modeling strategy such as that outlined by Blomberg 2017 BioRxiv <https://doi.org/10.1101/067363>.

RE: The reviewer is correct about how BM and OU models may not be ideal models of evolution for non-normally distributed traits. These evolutionary models are considered heuristic tools for describing and interpreting potential evolutionary patterns (Pennell and Harmon 2013) and fitting them makes sense in our comparative setting where rather than seeking a precise parameterization of thermal tolerance evolution we aim to establish which of three evolutionary scenarios is more likely: a) thermal tolerance variation is not structure across the phylogeny, b) thermal tolerance variation accumulates randomly with time, or c) thermal tolerance variation accumulates towards an “attractor”. Yet, as noted by the reviewer, there are no well-established alternatives to circumvent data skewness other than using simulations. To assess the adequacy of BM and OU models to fit skewed data such as ours, we now incorporate a simulation test. Its results suggest that thermal tolerance data is suited to discriminate differences between BM and OU models, regardless whether raw data or normalized data are used. That is, the differences we find between BM and OU models would still hold (and even be reinforced, see Figs. S3-S4) if data was normally distributed and thus, simulation results support our interpretation of an OU-type evolution of thermal tolerances.

Regarding the reviewer’s suggestion of an alternative modelling strategy, we consider that fitting Cox-Ingersoll-Ross or Beta evolution models as suggested by Blomberg (2017) to our data would have drawbacks because these models have not been empirically tested, and it is not clear what interpretations to make out of fitted parameters.

In our revised version of the MS, we acknowledge the limitations and assumptions of commonly-used models of evolution (see lines 341-411 supplementary material): “We acknowledge that BM and OU models, which are heuristic tools for describing and interpreting potential evolutionary patterns³⁶ may not be fully realistic models of trait evolution^{37,38}. However, by comparing these simplistic models we infer whether the evolution of thermal tolerance is best described by random increase of phenotypic variation in proportion to time or that phenotypic variation is subject to some sort of constraint (i.e. an OU-type evolution). We do not test for greater or lesser trait skew (one possible outcome of a strong physiological boundary in trait evolution), but instead we test for trait evolution to be closer to Brownian vs. slowing down around an “attractor” trait value. The latter would be consistent with several scenarios such as hard boundaries, strong trade-offs for thermal limits away from the attractor, or optimal fitness at the attractor.”

In addition, we deal with non-normality through a simulation test (see supplementary file lines 119-147 and Figs. S3 and S4).

3. For examining the climate legacies hypothesis, I think that more than glaciation status of the planet needs to be taken into consideration. Even during glaciations, some locals were warm and during interglacials some locals were cold. I think that this spatial variation needs to be accounted for in order to test climate legacies (e.g. paleo-biogeography rather than just current biogeography). I think that only using time is too coarse for testing this hypothesis.

RE: We agree with the reviewer, a hard test of the climate legacies hypothesis would require examining data on both the distribution of palaeoclimates and the palaeo-biogeography (i.e. palaeo-distributions of species and their ancestors). Unfortunately, such data do not exist at the deep-time, spatial and taxonomic scales used in our analysis and reconstructing them through a phylogeny requires many assumptions that are difficult to validate. Further, although coarse in nature, the aim of the analysis of palaeoclimate is to determine whether or not the (very roughly defined) climatic conditions at the origin of current species' ancestors have left a detectable imprint in their thermal tolerances, as predicted by the niche conservatism hypothesis. This is a simple but relatively strong test, and we find the expected signal. We expect that this signal would be only stronger if palaeo biogeographic data were available. In the revised version of the MS, we acknowledge this possibility (lines 352-359 supplementary information) "A more refined test of the climate legacies hypothesis would require examining data on both the distribution of palaeoclimates and the palaeo-biogeography (i.e. palaeo-distributions of species and their ancestors), unfortunately, such data do not exist at the deep-time. In general deep time estimation of climate is challenging and known to be associated with measurement errors8."

4. As written, each of the three hypotheses proposed to explain the evolution of thermal limits (LNS 72-84) seem overly simplistic. Moreover, they are pitched as alternative hypotheses, yet all are apparently supported by the data (LNS 48-51). I think that the paper would benefit from some rethinking of this framework with more nuanced hypotheses that better represent current knowledge in the field. For example, when considering hypothesis (1), even though taxa may have evolved in relatively "thermally stable" warm conditions, there is still sizeable variation present in the thermal environment resulting in organisms facing challenging temperatures. Current/recent environments present frequent selection where tolerance of higher temperatures would be beneficial. Thus hypothesis 1, as written, seems to require organisms never, or at least rarely, be subject to thermal challenge, which is not the case. For hypothesis (2), I agree that the literature indicates that upper thermal limits are less variable than lower thermal limits, but there is still variation which argues against a simple physiological barrier as described. Within animals alone, vertebrates seem to max out at ~46C (dessert iguana) although insects can commonly survive into the 50's C and the Pompeii worm can survive to ~80C. So "hard boundaries" as described already appears to be falsified. This is not to say that there would not be important evolutionary constraints that could be taxon specific or driven by phylogenetic inertia. However, this is a more nuanced hypothesis than what is presented. For hypothesis (3), the argument that "adaptation to rare extreme events" would explain the difference in variance between high and low temperature tolerance seems problematic because it ignores the difference in high and low temperature safety margins. Thermal performance curves are generally left-skewed resulting in smaller safety margins at high temperature. As a result, extreme warm events of less absolute magnitude can have larger effects on fitness than cold events of larger absolute magnitude. Also, the most important determinant of cold tolerance is frequently

freeze tolerance, which is not currently considered. All this to say that there are numerous a priori reasons to think that simple differences in the absolute magnitude of high and low-temperature thermal variation would not adequately explain the evolution of thermal tolerance.

RE: We agree with the reviewer that the competing mechanisms are not mutually exclusive. We thank the reviewer for drawing our attention to this ambiguity in the text and we have made revisions to clarify this point in line 95-97. “We analyse the Globtherm database²¹ to untangle the relative roles of these three (non mutually exclusive) mechanisms in shaping the global extent of variation in thermal physiological limits.”

Although performance curves are left skewed, we did not find evidence of stronger selection on heat tolerance as many others have also found (as pointed out by reviewer 1). However, we agree that this is an interesting discussion point that we have now incorporated in text (lines 71-76). Why selection has acted more on lower thermal tolerance, as per reviewer one’s suggestion, could be explained by the three hypotheses outlined, or behavioural heat buffering, all of which could reduce selective pressure on heat tolerance we have updated the text in lines 90-93 accordingly.

We also agree with the reviewer that some organisms have (exceptionally) evolved to live under extreme thermal conditions (incl. a restricted number of species among the phyla Pogonophora, Annelida, Mollusca and Arthropoda that have adapted to thermal regimes and extremes associated with hydrothermal vents, including the species *Alvinella pompejana*, as mentioned). These exceptions do not negate a finding that the majority of species become constrained at high upper thermal limits. We did not consider freeze tolerance because it is not a commonly measured metric across ectotherms and instead is only measured in a limited number of taxa. We agree with the reviewer that given there are exceptions to a “Hard physiological boundary” perhaps a better term is “strong physiological boundary” and have updated the text throughout.

Line Comments

LNS 78-80: This clause seems out of place/unrelated to the prior clause.

RE we have revised lines 80-84 for clarity: “*Strong physiological constraints could limit physiological adaptation beyond certain temperatures, for example, the inability of organisms to counter the destabilizing effects of high temperatures on membranes and proteins may lead to an optimum operating temperature i.e. ‘maxithermy’ hypothesis¹⁷. Adaptive and phenotypic responses are more often documented for lower thermal limits in animals and plants¹², which suggests that a boundary may have been reached by upper thermal limits¹².*”

LNS 95-96: This sentence is hard to follow as written, and seems to be missing a verb.

RE We have revised this sentence for clarity, revision lines 101-105: “To investigate species’ geographic and temporal distributions in relation to changes in climate at

evolutionary timescales, we documented (a) the ‘thermal ancestry’ of every species in our dataset, based on the palaeoclimatic conditions predominant on Earth at the time when its ancestor originated (see methods)¹⁵.

LNS 120-125: Although the observation that there is more variation in cold tolerance than heat tolerance and more taxa originated in “warm” time periods is consistent with the deep-time climate legacies hypothesis, I do not find it compelling. We knew a priori that cold temperature tolerance is more variable than high-temperature tolerance, and there were only two possible paleoclimate environments considered (i.e. warm or cold). Thus, there was a 50:50 chance of observing this. Also, it doesn’t account for the known plasticity of cold thermal tolerance.

RE: We acknowledge that our data has limitations, supplementary file lines 298-359. However, despite our data limitations our results provide novel insights and relevant directions for future work, as recognised by other reviewers. We respectively disagree with the reviewer about the limitations of using binary variables as our results show that the probability of the predictor being important is higher than random.

LNS 142-144: I think that this statement is too strong given that the only alternative models considered were Brownian motion and white noise. All that you can say is that the data were more consistent with an OU model. I’m not sure you can distinguish the difference between an optimum “attractor” and a wall however. I’m also not sure how well OU does with boundary conditions, given that this breaks one of the assumptions of the model (as described above).

RE: The reviewer is correct in that we can mainly only say that the data were more consistent with an OU model. Our idea was not to distinguish between an "optimum attractor" and a "wall", but simply to ask which of the two alternative models (BM or OU) better describe our data. We have revised to the manuscript to better reflect this point lines 149-152: “Variation in species’ thermal physiological limits across clades accumulated through time consistent with an Orstein-Uhlenbeck model of evolution, with fits for the α parameter suggesting a moderate to strong stabilizing selection ($-\log \alpha < 0$) towards an optimum value (Table S1).”

LNS 157-160: I don’t think that observed min and max temps are really a proxy for “extreme events” which are usually defined as being abnormal events (e.g. more than 2 sd from the mean). Looking at the max and min tell you nothing about the distribution of thermal conditions. Also, I cannot find details for the environmental max and min values used. Are these air temperatures or operative temperatures? Air temperatures can do a fairly good job of modelling min but are often quite different from the possible max experienced by organisms. Some additional description of these data would be useful.

RE: We acknowledge that these temperatures are used as proxies. However, the maximum and minimum temperatures experienced by an organism across the year in a given location does provide valuable information about the thermal conditions of a given place. Regarding the climate data, for terrestrial species it is air temperature and for marine it is sea surface temperature. We have revised the manuscript to provide more details on the climate data used lines 249-251: “Based on the

coordinates supplied in GlobTherm we extracted air temperature at 2.5 min resolution for terrestrial taxa³⁸ and sea surface temperature at 5 min resolution for marine taxa^{39,40}.”

LNS 207-208: I disagree that the analyses presented provide a mechanistic understanding of thermal evolution. We neither gain insight into the mechanisms driving selection nor the physiological processes most directly under selection/evolving. Rather, these analyses and their inferences seem correlative/statistical. For example, current and paleo-climates were correlated with thermal tolerance and variance. This is interesting, but I do not think it can accurately be called mechanistic.

RE lines 220-222: “Ultimately, our finding that thermal limits are constrained by evolution, and conserved through time across broad taxonomic groups, can inform predictions about how species would distribute under warmer or colder climates”

Reviewer #3 (Remarks to the Author):

This study provides a large-scale phylogenetic analysis of ‘thermal limits’ across the tree of life, and provides the same sort of message of previous reviews that heat tolerance somehow has reached a physiological boundary and may not be able to evolve in response to climate change. In this sense, the main take-home message is not entirely new. Nonetheless, one might still argue that analyses would be of interest to the general community if results were reliable, but unfortunately this is not the case. Analyses are oversimplistic and ignore a multitude of confounding factors that should be taken into consideration from the general quality of the data to the classification employed to reconstruct the paleoclimate associated with the origin of each evolutionary lineage.

Without going any further, there are at least two fundamental problems with the study in terms of the quality of the data: (1) thermal limit estimates vary substantially with measurement protocol and there are no attempts to control for confounding effects and (2) estimates of so-called ‘thermal limits’ between ectotherms and endotherms are by no means comparable.

With regards to the first point, there is extensive literature on the problems associated with the estimation of ‘thermal limits’ and their accuracy that spans more than a decade. To make a long story short, Lutterschmidt and Hutchison (1997) pointed out in the first review on the subject that critical endpoints were highly sensitive to differences in measurement protocol, and this problem was brought back to light following the empirical paper by Terblanche et al. (2007). Subsequently, Rezende et al. (2014) showed that, other confounding effects aside, previous comparative studies have ignored the impact of exposure time and that this variable accounts for nearly 64% of the variation across organisms/studies. This limitation remains in the current submission, thermal limit estimates are not directly comparable and consequently any sort of phylogenetic reconstruction that neglects the fact that we might be comparing apples and oranges will be biased and unreliable. Considering the diversity of data that is available at Globtherm, the bare minimum that is required here is to account for potential confounding effects such as (i) exposure time associated with the endpoint estimation, (ii) experimental protocol (ramping versus static) and (iii) developmental stage. For the second point, the authors pool estimates of thermal limits in ectotherms (i.e., endpoints where animals lose coordination, are knocked down or die; or what some

researchers call 'ecological death') with the limits of the thermoneutral zone (TNZ) in endotherms, which are simply not comparable. The limits of the TNZ correspond to the temperatures in which endotherms must elevate their metabolism above basal levels to thermoregulate at rest, and this is far from a thermal estimate of 'ecological death' for an endotherm. Accordingly, only a handful of endothermic species can apparently cope with temperatures below 0 °C in Fig 3b, while in reality many birds and mammals regularly encounter lower temperatures in their ranges of distribution. Yes, I understand that some studies have used TNZ limits as a standardized proxy to estimate thermal limits in endotherms, which is problematic but at least comparisons were performance across endothermic species. Here, the authors argue that these estimations are comparable to CTmax or other estimates of thermal limits in ectotherms, and this is simply wrong. TNZ limits have been adopted as a standard measure of thermal tolerance for endotherms for practical purposes because there are real estimations of CTmax or CTmin for birds and mammals, which is not to say that these estimates are in any way comparable to the endpoints where locomotion is disrupted or lethal temperatures. This explains, for instance, the counterintuitive result in Fig 3 that ectotherms can tolerate colder temperatures than endotherms when one of the main arguments to explain the evolution and diversity of endothermic lineages was to colonize cold environments and distribution patterns would suggest the opposite trend (see Grady et al 2019).

RE: We agree with the reviewer that different experimental measures are commonly associated with different groups of organisms. For this reason, we took steps to collate metrics that would be most comparable within groups, so that evolutionary rates among them would make sense. In plants and animals, these were exclusively dynamic ramping endpoints of critical organismal function (CT), and lethal limits under static conditions (LT), and we did not combine these with other definitions of thermal limits e.g. those based growth rate endpoints, or others based on specific physiological functions (such as heart pumping or respiration rates). We have updated the supplementary file lines 14-18 to clarify our reasoning and to make our methods clearer lines: "Different experimental measures are often associated with different groups of organism and in part, this is why we analysed endotherms (all measured using TNZ), plants and algae (almost exclusively lethal experiments), ectotherms (majority critical limits) separately (for more information on the phylogenetic analysis see below)".

The reviewer highlighted that there could be an issue with mixing lethal and critical measures in our analysis. To determine if this was the case, we analysed these groups separately were appropriate and included this new analysis in the supplementary material (Table S5-S6). We found that separating these groups did not change the interpretation of our results, therefore to maximise taxonomic coverage we have continued to discuss these groups together in text.

We agree with the reviewer with regards to the other aspects of the experimental design (i.e. ramping rate and acclimation temperature) that can affect the experimental endpoint. We understand the reviewer's concern and realize that we should have been more explicit about the influence that different experimental treatments can have in our results. This issue only affects estimates from ectotherms and to some extent plants (which are more commonly measured using static

measures). Endotherms use a standard experimental approach. With regards to ectotherms and plants we have now included in the supplementary material results from a Random Forest model that includes pretreatment and ramping rate as covariates. We found that although ramping rate and pre-treatment temperature are important in determining thermal tolerance, they did not change our conclusions qualitatively. We have updated the text (see lines: 234-238 and 309-318 supplementary file) and added Table S7 showing these results. We also cite a recent paper by some of the authors of this ms. where analyses explicitly explore the effects of experimental treatments (Sunday et al. 2019) and similarly found that ramping rate and pretreatment do not have a major effect on global patterns in thermal tolerance. Revision lines 310-319: “The Globtherm dataset was designed to be a database of comparable thermal tolerance metrics. To achieve it’s goal, preference was given to sources that used the most widely used methods and measurements and only included data from adults. However, differences in experimental design can affect the estimates of thermal end-point, this is particularly relevant to ectotherms when experiments use different ramping rates and pre-treatment temperatures. Data on ramping rate and pre-treatment temperature is available for a limited number of taxa in GlobTherm and including such treatments as covariates in our Random forests did not change our conclusions qualitatively (Table. S7). Our finding is consistent with previous studies which have shown the magnitude of the effect of ramping rate and pre-treatment temperature do not affect the interpretation of global patterns in thermal limits using the Globtherm dataset³¹.”

Ultimately, this study involves the use of some sophisticated phylogenetic statistical methods to reconstruct the evolution of thermal tolerance based on a global dataset that was already available, without any consideration regarding the quality of the data, what different estimates actually estimate, etc. Major sources of uncertainty underlying the distribution of thermal tolerance estimates have been completely dismissed, and therefore I am afraid analyses and results are simply not reliable.

RE: We thank the reviewer for noting our sophisticated statistical approach. We agree with the reviewer that although we had spent considered time developing our approach, which did consider the differences in experimental design used in ectotherms we did not explicitly discuss this in text. We thank the reviewer for drawing this to our attention and we have now updated the manuscript to address this oversight (see above response).

References

Felsenstein, J. (1988). Phylogenies and quantitative characters. *Annual Review of Ecology and Systematics*, 19(1), 445-471.

Grady et al. (2019) Metabolic asymmetry and the global diversity of marine predators. *Science* 25:363

Lutterschmidt, W.I. & Hutchison, V.H. (1997) *Canadian Journal of Zoology*, 75, 1553–1560.

Pennell, M. W., & Harmon, L. J. (2013). An integrative view of phylogenetic comparative methods: connections to population genetics, community ecology, and paleobiology. *Annals of the New York Academy of Sciences*, 1289(1), 90-105.

Rezende et al (2014) Tolerance landscapes in thermal ecology. *Funct Ecol* 28: 799-809.

Terblanche, J.S., Deere, J.A., Clusella-Trullas, S., Janion, C. & Chown, S.L. (2007) *Proceedings of the Royal Society B: Biological Sciences*, 274, 2935–2942.

Reviewers' Comments:

Reviewer #1:

Remarks to the Author:

Comments on how the authors responded to prior reviews:

On the issue that thermal limits of ectotherms are fundamentally different from thermal neutral zone of endotherms. All three reviewers were concerned with this. The current version is clear as to the differences here, but I personally feel the endotherm vs ectotherm data are too different to justify joint inclusion. The lower limit of endotherms will be dominated by body size (see Scholander's old work or Kearney & Porter's), and this does not hold for ectotherms. Further the upper critical limit for endotherms is driven by metabolic heat production exceeding evaporative cooling, such that T_b starts to rise

On whether paper-specific methodologies and metrics confound the analyses. Reviewers 2 & 3 were particularly concerned here. In the revision, analyses are done by group (endotherms, ectotherms, plants) helps. This helps but does not eliminate the problem. For example, consider Table S5 data for "CTMax" for ectotherms (for example): do the input data include both CTmax data and lethal temperatures? If I interpret Table S7 correctly, pretreatment and ramping rate do not obscure the base patterns. This is a useful addition.

Whether OU and BM models are appropriate (Reviewer 2). The objection here seems cogent, but the authors added a simulation test, which supports the base patterns.

Overall, I think they've done a good job of responding positively to the reviews (as I understand those).

REVISION

This paper takes an explicitly historical perspective on the evolution of tolerance limits on ectotherms, plants, and endotherms. That is a useful perspective to explore. But I come away somewhat frustrated and confused by the paper. For one thing, I'm unclear the take home. The Abstract (lines 51-54) implies that history plays a major role in cold tolerance limits. However, lines 163- 165 (and 169-170) state that current climatic "events" play the strong(est?) role in thermal tolerance variation. So there seems to be a disconnect between the Abstract and the Discussion. Perhaps this is forced by a word limit in the Abstract — but the inconsistency is confusing.

Some of the base findings (greater variation in CTmin than in CTmax) have been documented in many groups and thus are not novel. Granted, few studies have tried to look at rates on a tree; but stating that rates are greater in CTmin than in CTmax is inevitable, given the known patterns of intra-clade variation in those two traits.

One thing I don't understand. The authors look at contemporary climates, paleoclimates at the time of origin of a clade, and length of time since that origin. That is very interesting, but what about climate between the origin and the present? A lot has happened for many groups (Fig. 1), but there's no mention in the main text at least of the relevance of climate cycles after the origin. Imagine a clade that evolved in the Permian (cold) -- it will have experienced many climate cycles in the subsequent 250+ my. Might not the N and duration those cycles have an influence? Perhaps I'm missing something here, but I just think that if the "origin" has an impact, and if the present has an impact, than what happens between must have an impact as well.

I encourage the authors to very clear what is really novel here, and how that novelty I reinforces and

extends what we already know. I think the paper has the essence of that information, but it is very hard to retrieve from the text (I suspect this will especially so for those unfamiliar with the very long history of this field).

The writing adds to the complexity of this paper. Frequently I would read a sentence and wonder what it meant. I think the paper needs some serious editing before publication.

Below are some miscellaneous comments — many have to do with the writing style, which I find stilted and often unclear. I'll highlight examples from the first 100 lines. This is the key introduction to the paper, and I don't find it easy to read or understand (even after having read the paper several times).

In particular, I found the first paragraph hard to follow:

clades THAT originated
they may have lower cold tolerance limits but do they also have lower warm tolerance limits?
does heat limits relate to contemporary climates?

55 "in the relatively young endotherms" — at first I thought you meant young in chronological, 53 limits than those, that

evolved remarkably quickly relative to warm tolerance, to to rates for ectotherms?

57 "tolerances consistently showed slower evolution" -rephrase "heat tolerances evolved more slowly than cold tolerances, suggesting that an upper boundary for heat tolerance has been reached.

Very long introductory paragraph -- too long.
'On a palaeoclimatic scale, Earth has been predominantly warm with intermittent glaciations^{15,16}, such that deep-time climate legacies limit opportunities for taxa to evolve beyond the warm conditions during which most extant species and lineages arose¹⁷'

This is a key sentence, and I have trouble understanding what you mean. Are you saying that most living clades evolved during warm periods? I don't see the logical leap from the introductory phrase to the secondary one.

an optimum operational temperature ?
"most often documented" — ambiguous. You mean that lower limits show more variation than upper limits, but the wording could imply more studies have been done on lower limits.
I don't understand the link between the first phrase ("adaptation to survive...") and "given that..."
I think you are stating this as a null hypothesis, but I read this as your working hypothesis — so I was confused.
Is this indeed what you found? Where do you show that rarity determines tolerance limits?
"are being reached" —to me that implies an asymptotic pattern, which is not the evidence presented
I immediately wondered, which ancestor? Every species has more than one. Can you briefly explain how you chose ancestors, so readers don't have to search the Methods for this info?
Again a very long paragraph. I prefer several short paragraphs, as it is easy to get 'lost' in large

paragraph.

"current species geographic distribution." longitude/latitude/altitude, or did you compute some overall climate index for each species?

111... This is a key few sentences, but I have trouble understanding. A diagram might help. I can understand why you think both upper and lower limits might be lower than in a lineage that evolved during a warm period, but I don't follow why variability should differ. Some steps in your logic aren't apparent to me.

In any case are you assuming that if a clade evolved in a cold period say 3 glacials in the past, that its physiology is essentially stuck there through the following glacial and inter-glacial cycles. Why wouldn't apply to clades that evolved in warm periods

124 again I'm wondering what's a clade here? origin of the species, genus, family?

Interesting pattern. But rare the data ("analyzed species") biased, for example, by a hemisphere or climate difference in sampling? Recall that climate variability is generally larger in the N hemisphere

Addo-Bediako, A., S. L. Chown, and K. J. Gaston. 2000. Thermal tolerance, climatic variability and latitude. *Proceedings of the Royal Society of London, Series B* 267:739-745.

Fig. 1. The vast majority of clades of terrestrial ectotherms evolved > 200 million years ago. So "ancestor" is ancient ancestor!

Almost all the sampled plants evolved in warm period. Hard to believe that this is a balanced sample of plant origins

There's a curious white strip across the endotherm panels.

*** Here's where I'm having trouble. lots of terrestrial ectotherms evolved > 250 mya, and they experienced two modestly warm and one very warm (and long) period. Isn't that enough time at warm temperature to erase any signature of climate at the clades origination?

Fig. 2 Why does the x axis have two glaciated and two warms? Not explained in the legend.

Isn't this an expectation from a brownian motion model?
Role of contemporary climate?
an optimum(i.e., maxithermy) or a contrastint... If I remember maxithermy, Hamilton propose that warmer temperatures (for thermodynamic reason) enable higher physiological rates, but that selection could not push organisms to temperatures exceeding his maxithermy limit. "optimum" here seems confusing.
Interesting, but I think lower limits are more plastic (ontogenetically) than upper limits.
unclear — means that the endotherm data has an inordinate effect on the overall pattern, or that only endotherms evolved. "influenced by evolution in endotherm"
sure, but this is likely a size effect, as size has a huge effect on the lower critical limit of the thermal neutral zone.
this paragraph combines several themes and thus is hard to read. Splitting the paragraph would

help.

"rare and extreme climatic events" -what are your metrics here? Do you mean glacial interglacial cycles?
not the clearest of sentences.
"extreme climatic events" — again unclear. Do you min max environmental temperature? If so, that is NOT an EVENT to me — merely a range boundary. In any case, a few lines down we learn that you analyzed max and min temperatures separately. [I think so but "either measured through minimum and maximum environmental temperatures" is grammatically incorrect and unclear.]
OK, if contemporary Tmax and Tmin are the most important variables, that suggests that critical temperatures respond quickly to recent conditions. So why should age of clade matter? Does this seem inconsistent to you? Granted the pattern may be real, but it just seems inconsistent.
But if the max effect was 13%, doesn't seem like climate at the origin is relatively important.
Most readers won't understand what you are saying, which is that you included both CT and lethal limits, (even though these are different measurements with CT < lethal), as this increased the sample size.
Cite the prior literature, which for many taxa shows this pattern. Prior studies may not have measured rates per se, but if CTmax shows less variation in a clade than does CTmin, then CTmax must have lower rates of evolution.
Doesn't this contradict line 124?
optimum or maximum?
Not sure this finding applies, as these patterns apply to a genus, not to a major clade. In any case, knock-down experiments with *Drosophila* show rapid evolution of heat tolerance (either time to knockdown, or knockdown temperature).
tying this to oxygen limitation is a stretch. However, let's assume it is involved (a Pörtner "effect"). Then if Berner is correct as to major variation in atmospheric O₂ levels over geological time, then look at clade origins in the Permian (cold, high O₂) vs those in the Triassic (hot, low O₂). Can you separate temperature from O₂ effects?

219 Did Brattstrom state that? In any case, add a few words to explain the logic.

Discussion — some of the ideas here (thermal limits are constrained and conserved) have been around for decades

Reviewer #2:

Remarks to the Author:

I appreciate this thorough revision which appears to resolve my greatest concerns with the previous version of the MS. In particular, I appreciate the thorough simulations of the evolutionary models, division of analyses such that a single measure of thermal tolerance is used within each analysis, the implementation of Random Forest models for assessing the relative importance of each factor tested, and tests for the effect of methodology on conclusions.

I still have more-minor concerns, although I think these can be readily addressed by the authors. In particular, I think that the primary MS is missing important nuance which results in overstating some results and their implications. I worry that the vast majority of readers will not consult the supplement and thus leave the MS with potentially the wrong idea. More specific details are in my line comments below.

Line Comments

Line 148 and 155: I think Table S1 should be moved to the primary MS. It provides the primary results for two of the most compelling conclusions of the paper: that tempo differs between upper and lower thermal tolerance, and that the OU model is better supported by the data than BM or WN models. I don't think readers can really appreciate these results without referencing this information.

Additionally, this is the one spot where analyses do not appear to be divided such that different measures of thermal tolerance are not included in the same analysis. I think this division is absolutely necessary to draw conclusions. As far as I can tell, the "All Species" scenario fit OU, BM, and WN models to all of the species simultaneously regardless of the measure of thermal tolerance employed. I think that this equates to testing an evolutionary model where both apple and orange phenotypes are included but the data are interpreted as if all were apples. This will affect interpretation even more given the strong covariation between phylogeny and measurement in this dataset. I think that the "All Species" scenario is invalid and should be dropped and statistics from this analysis should not be used in the primary MS to support conclusions, such as on lns 146-147. Rather, each taxonomic/thermy grouping should be referenced individually for drawing conclusions.

Lines 153-158: I think additional care should be taken when describing the results and conclusions from the comparison of the evolutionary models. For example, I disagree that the results indicate that "limits across clades accumulated consistent with an Ornstein-Uhlenbeck model of evolution" because I don't think the model selection methodology can allow such a conclusion. Rather, the tests indicate evolution of the limits was more consistent with OU than the alternative null models of BM or WN. I think that this is an important distinction to be clear about, especially when potentially communicating to a very broad readership.

Additionally, I think additional care should be taken describing interpretations of the alpha parameter. In different spots in the MS this is interpreted as evidence for "moderate to strong stabilizing selection" as well as evidence for "strong physiological boundaries". However, I think that these are really different things. Stabilizing selection toward an optimum implies that phenotypes on either side of the optimum are possible and occur but result in reduced fitness. Thus, the value that we observe would be the optimum providing the highest relative fitness. By contrast, a strong physiological boundary would result in higher values not being possible. Strong directional selection could then move the phenotype right up to the boundary but not be able to take it further because higher values never occur to be selected for.

Because you only included one model of selection (OU) and two model of no selection (BM and WN), I think you can conclude that data are consistent with selection and/or constraints shaping the critical thermal limits, but cannot imply that the data are indicative of stabilizing selection. My preference would be to be as clear about the model as possible. The OU model being supported over the others most technically indicates evidence for phenotype values that act as "attractors," rather than the phenotype evolving at random. You could then go on to explain that the attractor could result from stabilizing selection, but given the other observations directional selection followed by a physiological barrier may be the more parsimonious conclusion. Again, because this is targeted to a general audience that may not be familiar with the ins-and-outs of evolutionary models, I think caution is warranted in describing what the tests really mean.

Line 161 and 162. I think that this is an interpretation that does not belong in the results section. The

models did not directly examine "adaptation to survive rare and extreme current climatic events" but instead looked at effects of clade age, paleo temperature, and current temperatures.

Line 201. Similar to my concerns described two comments previously. I think this statement should be softened. For example, "attractor" could be a more appropriate term than "optimum".

Reviewer #1 (Remarks to the Author):

Comments on how the authors responded to prior reviews:

On the issue that thermal limits of ectotherms are fundamentally different from thermal neutral zone of endotherms. All three reviewers were concerned with this. The current version is clear as to the differences here, but I personally feel the endotherm vs ectotherm data are too different to justify joint inclusion. The lower limit of endotherms will be dominated by body size (see Scholander's old work or Kearney & Porter's), and this does not hold for ectotherms. Further the upper critical limit for endotherms is driven by metabolic heat production exceeding evaporative cooling, such that T_b starts to rise

On whether paper-specific methodologies and metrics confound the analyses. Reviewers 2 & 3 were particularly concerned here. In the revision, analyses are done by group (endotherms, ectotherms, plants) helps. This helps but does not eliminate the problem. For example, consider Table S5 data for "CTMax" for ectotherms (for example): do the input data include both CTmax data and lethal temperatures? If I interpret Table S7 correctly, pretreatment and ramping rate do not obscure the base patterns. This is a useful addition.

Whether OU and BM models are appropriate (Reviewer 2). The objection here seems cogent, but the authors added a simulation test, which supports the base patterns.

Overall, I think they've done a good job of responding positively to the reviews (as I understand those).

RE: We appreciate the acknowledgement of our previous effort to address concerns raised by reviewers in a previous round of reviews. We have worked further to address new comments, refine any remaining methodological issues and polish the text to clarify any aspects that were not sufficiently clear. This includes figure captions and table legends. For example, we are now clear in Table S5 that each line presents detailed information for a single metric in each group (e.g. CTmax or CTmin only for ectotherms, no lethal temperatures included). In addition, we have reconsidered the joint inclusion of all species to finally dismiss such results, following reviewers' concerns. We also discuss the possible role of body size on the lower limit of endotherms, but not for ectotherms based on the mechanistic findings by Porter & Kearney (2009) or Rubalcaba & Olalla-Tarraga (2020). Please see below our response to the reviewers with details about the changes made to this revised version of our ms.

REVISION

This paper takes an explicitly historical perspective on the evolution of tolerance limits on ectotherms, plants, and endotherms. That is a useful perspective to explore. But I come away somewhat frustrated and confused by the paper. For one thing, I'm unclear the take home. The Abstract (lines 51-54) implies that history plays a major role in cold tolerance limits. However, lines 163- 165 (and 169-170) state that current climatic "events" play the strong(est?) role in thermal tolerance variation. So there seems to be a disconnect between

the Abstract and the Discussion. Perhaps this is forced by a word limit in the Abstract — but the inconsistency is confusing.

RE: We thank the Reviewer for pointing out that they find our manuscript explicit historical perspective on the evolution of tolerance limits on ectotherms, plants, and endotherms a useful one. We are also grateful that the Reviewer has provided us with the opportunity to revise the abstract. We have now attentively rephrased the abstract to more clearly convey the message that both (1) adaptation to current climatic extremes and (2) evolutionary constraints on physiological boundaries, appear to account for most of the extant observed variation in thermal tolerance limits across the tree of life.

Some of the base findings (greater variation in CT_{min} than in CT_{max}) have been documented in many groups and thus are not novel. Granted, few studies have tried to look at rates on a tree; but stating that rates are greater in CT_{min} than in CT_{max} is inevitable, given the known patterns of intra-clade variation in those two traits.

RE: We agree with the Reviewer that the novelty of our work does not rely in documenting greater variation for tolerance to cold than for tolerance to heat. This said, we would argue that there are several novel aspects of our study, including its scope (i.e. the largest database analysed to date in terms of taxonomic and geographic scope), its phylogenetically explicit analyses, and its conceptual framework (i.e. we test three specific hypotheses aimed at explaining thermal tolerance variation). It is worth mentioning the explicit novelty in documenting patterns for the tempo and mode of evolution for thermal tolerances across clades, metrics and taxonomic levels. We have made an effort in our revision to be specific on all aspects of novelty our MS bring forward.

One thing I don't understand. The authors look at contemporary climates, paleoclimates at the time of origin of a clade, and length of time since that origin. That is very interesting, but what about climate between the origin and the present? A lot has happened for many groups (Fig. 1), but there's no mention in the main text at least of the relevance of climate cycles after the origin. Imagine a clade that evolved in the Permian (cold) -- it will have experienced many climate cycles in the subsequent 250+ my. Might not the N and duration those cycles have an influence? Perhaps I'm missing something here, but I just think that if the "origin" has an impact, and if the present has an impact, than what happens between must have an impact as well.

RE: The reviewer is correct. We agree that variations in climate between the clade origin and the present could potentially exert selective pressure on thermal tolerance. Indeed, in section 5 of the Supplementary Information, which discusses caveats and limitations of data and analyses, we assume that a more refined test of the *deep-time climate legacies* hypothesis would require examining data on both the distribution of palaeoclimates and the palaeo-biogeography (i.e. palaeo-distributions of species and their ancestors). Unfortunately such data do not exist to our knowledge, although some studies have simulated these changes at the continental scale (Rangel et al. 2018:Nature). Even so, one of the interesting results that arises from our analyses is that the effects of contemporary climatic conditions do not mask fully the emergence of deep-time climatic signals and, especially, the relevance of evolutionary constraints on heat tolerance, as the more physiologically bounded thermal limit in ectothermic and endothermic organisms: according to the tempo of evolution. In ectotherms, the imprint of ancestral climatic affinities is still evident on cold tolerance. While prevailing palaeoclimatic conditions at the origin of a clade are unrelated to the evolution of heat tolerance, species that originated at a time of partial or full glaciations in the Earth's

history tolerate colder temperatures than species of warm thermal ancestry. This is now highlighted in lines 138-143 and implies that at least for cold tolerance what happens at clade origin appears to leave a solid imprint that was not erased by successive climate cycles.

I encourage the authors to very clear what is really novel here, and how that novelty I reinforces and extends what we already know. I think the paper has the essence of that information, but it is very hard to retrieve from the text (I suspect this will especially so for those unfamiliar with the very long history of this field).

RE: We are grateful for the Reviewer's encouragements in more clearly stressing the novelty and major advances our work generates. As described above, we have revised every section of the MS in an effort to better clarify how it adds scientific knowledge. In the introduction we use the first paragraph to cite and present background information on the topic we address. We have also paid special attention to better describe and introduce the rationale for each of the hypotheses we are testing.

The writing adds to the complexity of this paper. Frequently I would read a sentence and wonder what it meant. I think the paper needs some serious editing before publication.

RE: We thank the reviewer for this and have thoroughly edited the manuscript for clarity throughout.

Below are some miscellaneous comments — many have to do with the writing style, which I find stilted and often unclear. I'll highlight examples from the first 100 lines. This is the key introduction to the paper, and I don't find it easy to read or understand (even after having read the paper several times).

In particular, I found the first paragraph hard to follow:

52 clades THAT originated

RE: Amended.

they may have lower cold tolerance limits, but do they also have lower warm tolerance limits?

RE: This hypothesis has been clarified in lines 80-89.

does heat limits relate to contemporary climates?

RE: To specifically answer this question, yes, our results support that both heat and cold tolerances relate to contemporary climate. This is now articulated clearly in the MS. In general, we have better presented the information in the abstract to stress the importance of contemporary climate for thermal limits.

55 "in the relatively young endotherms" — at first I thought you meant young in chronological,

RE: We have replaced with "in the more recently originated endotherms".

53 limits than those, that

RE: This sentence was deleted after rewriting the abstract.

evolved remarkably quickly relative to warm tolerance, to to rates for ectotherms?

RE: Indeed, we have clarified that cold tolerance in endotherms has evolved remarkably quickly “compared to ectotherms and plants” (LN 56-58).

57 “tolerances consistently showed slower evolution” -rephrase “heat tolerances evolved more slowly than cold tolerances, suggesting that an upper boundary for heat tolerance has been reached.

RE: This sentence is no longer in the abstract.

Very long introductory paragraph -- too long.

RE: Indeed, the introduction has now been broken into different paragraphs to help it read more fluently, with additional text changes and rewriting to accommodate the suggestions made below by the reviewer.

‘On a palaeoclimatic scale, Earth has been predominantly warm with intermittent glaciations^{15,16}, such that deep-time climate legacies limit opportunities for taxa to evolve beyond the warm conditions during which most extant species and lineages arose¹⁷’

This is a key sentence, and I have trouble understanding what you mean. Are you saying that most living clades evolved during warm periods? I don’t see the logical leap from the introductory phrase to the secondary one.

RE: We have revised this part to better introduce the *deep-time legacies* hypothesis (see LN 80-89).

an optimum operational temperature ?

RE: This sentence has been corrected, as there was a terminological misuse. We meant “constrain variation in heat tolerance” and we have further clarify this hypothesis (see LN 91-106).

“most often documented” — ambiguous. You mean that lower limits show more variation than upper limits, but the wording could imply more studies have been done on lower limits.

RE: Indeed, we have revised this sentence to avoid confusion. It now reads “Lower thermal limits have been documented to exhibit more variation than upper thermal limits in both animals and plants”

I don’t understand the link between the first phrase (“adaptation to survive...”) and “given that...”

RE: We have replaced the name for this hypothesis, which could generate confusion. After revision it now reads (line 108):

“Third, adaptation to current climatic extremes is expected to exert selective pressure on thermal limits. However, we expect current climatic extremes to exert greater selection on lower compared to upper thermal limits for two reasons. On one hand, maximum environmental temperatures tend to be less variable across contemporary biogeographic gradients (i.e. latitude) compared to minimum temperatures. On the other hand, behavioural buffering is more likely to reduce selective pressure on heat tolerance relative to cold tolerance, because while organisms are able to use behaviour to evade heat stress, there tends to be fewer opportunities to behaviourally evade cold stress”

I think you are stating this as a null hypothesis, but I read this as your working hypothesis — so I was confused.

RE: This is indeed one of the three hypotheses we are testing. We have rewritten this section for clarity, which now reads “If adaptation to current climatic extremes is the main determinant of species’ thermal tolerance limits, we would expect a close match between experienced and tolerated temperature extremes and similar rates of evolution in upper and lower thermal limits, with no optimum or boundary.”

Is this indeed what you found? Where do you show that rarity determines tolerance limits?

RE: We have removed the term “rare” here and elsewhere as it perhaps does not best describe our analysis, which is focused on extreme temperatures and not rare climatic events.

“are being reached” —to me that implies an asymptotic pattern, which is not the evidence presented

RE: This sentence has been deleted.

I immediately wondered, which ancestor? Every species has more than one. Can you briefly explain how you chose ancestors, so readers don’t have to search the Methods for this info?

RE: We are now more explicit about our focus on the order taxonomic level, as a proxy for ancestry (see line 132-142 and 312-322). The reviewer is right; by definition any species would have multiple ancestors: according to where the temporal or taxonomic threshold is set. Our choice was not arbitrary (please see a detailed justification in the Methods section), instead it allowed a sensible trade-off between the variation in palaeoclimatic conditions assigned to each taxon and the number of taxa. Nevertheless, we conducted sensitivity analyses showing that our main results are not affected by the choice of taxonomic level (see also SI).

Again a very long paragraph. I prefer several short paragraphs, as it is easy to get ‘lost’ in large paragraph.

RE: To improve clarity, this paragraph has been split into three sections.

“current species geographic distribution.” longitude/latitude/altitude, or did you compute some overall climate index for each species?

RE: We thank the Reviewer to point this out, as it was indeed unclear. We used both minimum and maximum temperatures recorded at the location where the specimens of each species with data in the GlobTherm dataset were collected. We have now clarified this in several parts of the text (e.g. LN 136) and the Supplementary Information.

111... This is a key few sentences, but I have trouble understanding. A diagram might help. I can understand why you think both upper and lower limits might be lower than in a lineage that evolved during a warm period, but I don’t follow why variability should differ. Some steps in your logic aren’t apparent to me.

RE: We have rephrased the text with no mention to differences in variability (LN 85-89). We feel this new way to present the hypothesis improves clarity.

In any case are you assuming that if a clade evolved in a cold period say 3 glacials in the past, that its physiology is essentially stuck there through the following glacial and inter-glacial cycles. Why wouldn't apply to clades that evolved in warm periods

RE: We hope the simplified logic behind the *deep-time climate legacies* hypothesis, with no implicit assumptions on differential variability for thermal limits depending on climatic origin, is now better presented in lines 80-89. We are explicit on the importance of the interaction between niche conservatism (i.e. the trend to retain ancestral climatic affinities through time) and the Earth's climate history for this hypothesis.

again I'm wondering what's a clade here? origin of the species, genus, family?

RE: We have clarified that we focused on the Order level analysis in the main text. Please see above our response to point 108 for a revised version of the text.

Interesting pattern. But rare the data ("analyzed species") biased, for example, by a hemisphere or climate difference in sampling? Recall that climate variability is generally larger in the N hemisphere

Addo-Bediako, A., S. L. Chown, and K. J. Gaston. 2000. Thermal tolerance, climatic variability and latitude. *Proceedings of the Royal Society of London, Series B* 267:739-745.

RE: Our data are distributed relatively equally across both hemispheres (see Fig. 1). Specifically, UTLs are 60 % Northern hemisphere and 40 % Southern, while estimates for LTLs are approximately 50-50 split between hemispheres. However, most species with a cold ancestral origin are in the N hemisphere likely because the continents of the S hemisphere have mostly experienced warm palaeoclimates. In sum, the pattern is not likely to be due to differential sampling across continents but is more likely due to a real biological pattern that underlies the reported geographic distribution. We discuss this in the MS clearly in LN 146-156.

Fig. 1. The vast majority of clades of terrestrial ectotherms evolved > 200 million years ago. So "ancestor" is ancient ancestor!

RE: Ancestry is defined on the basis of the orders to which extant species belong. Clearly, the species we analyse are much more recent (please see detailed justification for this choice in section 1 of the Supporting Information). We have edited the figure's caption to ensure clarity.

Almost all the sampled plants evolved in warm period. Hard to believe that this is a balanced sample of plant origins

RE: The analysed dataset, Globtherm, contains approximately 250 plant species (not including algae) for which experimental estimates of upper and/or lower thermal tolerances are available. These species represent ~40 Orders. The age of order origin ranges from 66-436 mya, with a medium order age of 136 mya. As the reviewer points out, our dataset comprises a rather limited representation of all plant extant Orders for which thermal tolerance data are available. Thus, our results must be interpreted with caution as they may exclusively apply to the taxa analysed. However, it will be interesting to see if the patterns we report will hold as data for more taxa become available. We are now clearer about this point in the Caveats section of the Supporting Information.

There's a curious white strip across the endotherm panels.

RE: The white line is a gap in the axis. This was done for endotherms only, so that patterns in upper and lower thermal limits could be visualized separately for this group (note split in y-axis). The thermal neutral zone, which is only measured in endotherms in Globtherm produces a much narrower range between the upper and lower boundary, which caused an overlap of lower and upper limits in the figure. The figure caption has been updated to explain this, now reading: "For endotherms only, the axis is broken so that upper and lower thermal limits can be clearly delineated."

*** Here's where I'm having trouble. lots of terrestrial ectotherms evolved > 250 mya, and they experienced two modestly warm and one very warm (and long) period. Isn't that enough time at warm temperature to erase any signature of climate at the clades origination?

RE: This is precisely the motivation to explore one of the proposed hypotheses: the '*deep-time climate legacies*'. Consistent with the literature on niche conservatism, this hypothesis predicts that clades would tend to retain ancestral climate affinities in spite of the long evolutionary times spanned since ancestors emerged. One might expect that 250 mya – including very long periods of warm climates – should have erased any climate signatures for clades that emerged under cold periods. However, our results for ectotherms show that species belonging to clades that arose under cold climates tolerate colder temperatures than species from warm-emerged clades (see Fig. 2A). While our results show support for this hypothesis, our models that account for multiple factors simultaneously indicate that the importance of paleoclimate is rather limited when compared to that of other factors. Altogether, our analyses and results comprehensively respond to the Reviewer's concern. We now stress with the greatest possible clarity this finding, whilst putting them in context, in the main text (see Lines 121-132 or 218-227).

Fig. 2 Why does the x axis have two glaciated and two warm? Not explained in the legend.

RE: The figure caption has been revised for clarity

Isn't this an expectation from a brownian motion model?

RE: this sentence has been deleted in the revised version.

Role of contemporary climate?

RE: We are now clearer that both contemporary climate and evolutionary constraints on physiological boundaries can better account for the observed variation in heat tolerance.

an optimum(i.e., maxithermy) or a contrast... If I remember maxithermy, Hamilton propose that warmer temperatures (for thermodynamic reason) enable higher physiological rates, but that selection could not push organisms to temperatures exceeding his maxithermy limit. "optimum" here seems confusing.

RE: We agree that "optimum" is not a proper terminology and have revised this sentence (LN 102-106).

Interesting, but I think lower limits are more plastic (ontogenetically) than upper limits.

RE: Following the suggestion of Ref#2, we have deleted this sentence since the assertion that lower thermal limits have evolved over 3.5 times faster than upper thermal limits dealt with a comparison for "All-species" that is no longer reported (see response to comment by Ref#2 below).

unclear — means that the endotherm data has an inordinate effect on the overall pattern, or that only endotherms evolved. “influenced by evolution in endotherm”

RE: This comparison for “All-species” is no longer included, following the suggestion of Ref#2. Table 1 shows the tempo and mode for each of the three groups which all show the same pattern of faster evolution in lower thermal limits. Among the three groups, endotherms showed the fastest rates (Table 1). We have clarified this in the text.

sure, but this is likely a size effect, as size has a huge effect on the lower critical limit of the thermal neutral zone.

RE: We have now included a sentence to accommodate this possibility (LN 176-181): “The differences are markedly larger for endotherms than for ectotherms or plants (Fig. 3A-C; see also Fig. S2), which possibly reflects the different determinants of thermal neutral zones of endotherms compared to thermal limits of ectotherms and plants. For instance, biophysical models have documented an effect of increasing body size and decreasing fur insulation on the lower limits of the TNZ in mammals (Porter & Kearney 2009), but no relationship of size on CT_{min} in lizards (Rubalcaba & Olalla-Tarraga, 2020).”

this paragraph combines several themes and thus is hard to read. Splitting the paragraph would help.

RE: For clarity the paragraph has now been split into three parts.

“rare and extreme climatic events” -what are your metrics here? Do you mean glacial interglacial cycles?

RE: We apologize for the confusion. We now explicitly refer to “current climatic extremes” as we examine maximum and minimum environmental temperatures contemporarily experienced by species (see answer above). We have revised the manuscript throughout to clarify that we are referring to “current climate” and avoid the words “rare” and “events”.

not the clearest of sentences.

RE: This paragraph has been fully rewritten for clarity. In addition, further details on data collection and analyses are reported as Supplementary Information.

“extreme climatic events’ — again unclear. Do you min max environmental temperature? If so, that is NOT an EVENT to me — merely a range boundary. In any case, a few lines down we learn that you analyzed max and min temperatures separately. [I think so but “either measured through minimum and maximum environmental temperatures” is grammatically incorrect and unclear.]

RE: The reviewer is correct. Throughout the text we now refer to this as the “current climatic extremes” hypothesis and do not mention the terms “rare” or “event,” which we are not directly measuring. We have rephrased this sentence for clarity.

OK, if contemporary T_{max} and T_{min} are the most important variables, that suggests that critical temperatures respond quickly to recent conditions. So why should age of clade matter? Does this seem inconsistent to you? Granted the pattern may be real, but it just seems inconsistent.

RE: That contemporary T_{max} and T_{min} are the most important variables and they do not preclude the additional explored mechanisms to play a role. Actually, the hypotheses that we

explore are explicitly mentioned not to be mutually exclusive. In other words, they offer complementary mechanisms. More importantly, according to our results, clade age may be similarly important to current temperature for ectotherms and endotherms (see Fig. 4). To address this concern, we have edited the main text to stress that our hypotheses are not mutually exclusive (LN 78).

But if the max effect was 13%, doesn't seem like climate at the origin is relatively important.

RE: We agree and have revised the text accordingly (LN 217-225).

Most readers won't understand what you are saying, which is that you included both CT and lethal limits, (even though these are different measurements with $CT < lethal$), as this increased the sample size.

RE: We have revised this sentence (see LN 324-327), which now reads "To increase taxonomic coverage and sample size in ectotherms and plants, here we present results using both lethal and critical thermal limits: although limits of thermal neutral zones were exclusively analysed in endotherms."

Cite the prior literature, which for many taxa shows this pattern. Prior studies may not have measured rates per se, but if CT_{max} shows less variation in a clade than does CT_{min} , then CT_{max} must have lower rates of evolution.

RE: We have deleted this line in the new version of the text.

Doesn't this contradict line 124?

RE: The sentence contained in line 124 did not refer to an explicit test of a BM evolutionary model but mentioned the pattern depicted in fig. 1b regarding clade age. We now pay special attention to discuss the importance and specificities of an Orstein-Uhlenbeck (OU) model of evolution so that the reader better interprets its significance compared to BM and White Noise.

optimum or maximum?

RE: "optimum" has been replaced with "attractor" following suggestion by ref#2.

Not sure this finding applies, as these patterns apply to a genus, not to a major clade. In any case, knock-down experiments with *Drosophila* show rapid evolution of heat tolerance (either time to knockdown, or knockdown temperature).

RE: It is true that the pattern applies only to a genus and may not extrapolate to a more major clade, however, we need to interpret our results in light of previous literature where evidence exists. We now provide a reference in support to this sentence, as we were referring to findings by Kellerman et al. (2012, PNAS).

tying this to oxygen limitation is a stretch. However, let's assume it is involved (a Pörtner "effect"). Then if Berner is correct as to major variation in atmospheric O_2 levels over geological time, then look at clade origins in the Permian (cold, high O_2) vs those in the Triassic (hot, low O_2). Can you separate temperature from O_2 effects?

RE: Although beyond the scope of this study, we agree it would be interesting to test this idea in future studies to try and tease apart temperature from O_2 effects. Ultimately, whether or not temperature effects can be disentangled from O_2 effects will depend on data quality

and availability. For example, a possible test could restrict analysis to aquatic insects even though the OCLTT idea is being strongly challenged these days.

219 Did Brattstrom state that? In any case, add a few words to explain the logic.

RE: We have added an explanation with more appropriate citations, which now reads (LN 256-258):

“Because the majority of species evolved in warm environments, the species poor and cooler higher latitudes offer opportunities for speciation and evolution of thermal tolerance through adaptive radiation^{13,32}.”

Discussion — some of the ideas here (thermal limits are constrained and conserved) have been around for decades

Re: We have revised the text in order to more clearly acknowledge historical work and to put a greater emphasis on the specific novelty of our work.

Reviewer #2 (Remarks to the Author):

I appreciate this thorough revision which appears to resolve my greatest concerns with the previous version of the MS. In particular, I appreciate the thorough simulations of the evolutionary models, division of analyses such that a single measure of thermal tolerance is used within each analysis, the implementation of Random Forest models for assessing the relative importance of each factor tested, and tests for the effect of methodology on conclusions.

Re: We thank the reviewer for appreciating our revision and analyses. Please see below detailed responses to comments and concerns.

I still have more-minor concerns, although I think these can be readily addressed by the authors. In particular, I think that the primary MS is missing important nuance which results in overstating some results and their implications. I worry that the vast majority of readers will not consult the supplement and thus leave the MS with potentially the wrong idea. More specific details are in my line comments below.

Re: We agree that some of our results would have risked remaining buried in the SI preventing readers from a fairer evaluation of their reach. We have downplayed results that may have sounded overstated and addressed other concerns. Please see below details on how.

Line Comments

Line 148 and 155: I think Table S1 should be moved to the primary MS. It provides the primary results for two of the most compelling conclusions of the paper: that tempo differs between upper and lower thermal tolerance, and that the OU model is better supported by the data than BM or WN models. I don't think readers can really appreciate these results without referencing this information.

RE: It was our intent to convey such a message through Figure 3, but we realize more detail should accompany the main text. Thus, Table S1, which shows the Tempo and mode of evolution of upper and lower thermal tolerance, is now Table 1 in the main text and is referenced appropriately (in LN 172, 185, 188, 193, 196, 237, and 239).

Additionally, this is the one spot where analyses do not appear to be divided such that different measures of thermal tolerance are not included in the same analysis. I think this division is absolutely necessary to draw conclusions. As far as I can tell, the “All Species” scenario fit OU, BM, and WN models to all of the species simultaneously regardless of the measure of thermal tolerance employed. I think that this equates to testing an evolutionary model where both apple and orange phenotypes are included but the data are interpreted as if all were apples. This will affect interpretation even more given the strong covariation between phylogeny and measurement in this dataset. I think that the “All Species” scenario is invalid and should be dropped and statistics from this analysis should not be used in the primary MS to support conclusions, such as on lns 146-147. Rather, each taxonomic/thermy grouping should be referenced individually for drawing conclusions.

RE: We understand and agree with the reviewer’s concern; we should have been clearer about the aim of an all-species analysis. Beyond our aim to provide an accurate estimate of mode and tempo of evolution of thermal tolerances across the tree of life, the all-species analysis had been used as a confirmatory analysis to address whether or not patterns for different clades would hold after data aggregation. However, we realize now that, whilst our intention was that to emphasize the caveats of interpreting such a result, our wording could easily lead to think we were over- or misinterpreting its scope. Thus, we have now removed all analyses that aggregate “all species” in this way. Specifically, we have (1) removed the all-species results from previous Table S1 (now Table 1 in the main text), (2) modified previous Table S2 to show results for higher taxonomic levels (i.e. order and family) for the same groupings considered in the main text (now Table S1 in the Supporting Information), and (3) removed mentions to the all-species result from the text.

Lines 153-158: I think additional care should be taken when describing the results and conclusions from the comparison of the evolutionary models. For example, I disagree that the results indicate that “limits across clades accumulated consistent with an Ornstein-Uhlenbeck model of evolution” because I don’t think the model selection methodology can allow such a conclusion. Rather, the tests indicate evolution of the limits was more consistent with OU than the alternative null models of BM or WN. I think that this is an important distinction to be clear about, especially when potentially communicating to a very broad readership.

RE: We thank the reviewer for raising this important point. We have now edited the text in LN 183, which now reads:

“Variation in species’ thermal physiological limits across clades accumulated through time in a more consistent manner with an Ornstein-Uhlenbeck (OU) model of evolution than with the alternative Brownian Motion or White Noise models (Table 1; see also Phylogenetic sensitivity analyses and Fig. S2 in the Supporting Information).”

Additionally, I think additional care should be taken describing interpretations of the alpha parameter. In different spots in the MS this is interpreted as evidence for “moderate to strong stabilizing selection” as well as evidence for “strong physiological boundaries”. However, I think that these are really different things. Stabilizing selection toward an optimum implies that phenotypes on either side of the optimum are possible and occur but result in reduced fitness. Thus, the value that we observe would be the optimum providing the highest relative fitness. By contrast, a strong physiological boundary would result in higher values not being possible. Strong directional selection could then move the phenotype right up to the boundary but not be able to take it further because higher values never occur to be selected

for. Because you only included one model of selection (OU) and two model of no selection (BM and WN), I think you can conclude that data are consistent with selection and/or constraints shaping the critical thermal limits, but cannot imply that the data are indicative of stabilizing selection. My preference would be to be as clear about the model as possible. The OU model being supported over the others most technically indicates evidence for phenotype values that act as “attractors,” rather than the phenotype evolving at random. You could then go on to explain that the attractor could result from stabilizing selection, but given the other observations directional selection followed by a physiological barrier may be the more parsimonious conclusion. Again, because this is targeted to a general audience that may not be familiar with the ins-and-outs of evolutionary models, I think caution is warranted in describing what the tests really mean.

RE: The reviewer raises an important point, which was a source of debate amongst co-authors on previous versions of the MS. The interpretation of OU models is rather contentious, and since their original use in population genetics to support evidence for stabilizing selection in adaptive landscapes (Lande 1976, *Evolution*), these models have been imported in macroevolutionary studies, interpreted (and sometimes misinterpreted) as evidence for stabilizing selection, for phylogenetic niche conservatism, or even for convergent evolution (Cooper et al. 2016, *Biological Journal of the Linnean Society*). We had followed some of that literature to interpret our results, but now realize doing so could lead to confusion. Nevertheless, in the context of our hypotheses, we would argue that support for an OU model suggests that (1) the evolution of thermal tolerance is not random (i.e. WN) and cannot be explained through simple accumulation of phenotypic variation with time (i.e. BM), and (2) phenotypic values seem to cluster around ‘attractor’ values, which would agree with a loose definition of boundary (i.e. a value beyond which only limited phenotypic variation accumulates). We agree with the reviewer that such a ‘loose’ definition conflicts with our admittedly overplayed term “strong physiological boundaries”, and thus now refer to it as “physiological boundaries”. Further, we have changed the wording of this result, replacing “optimum” with “attractor”, and specifically call for caution regarding its interpretation, as inference of evolutionary processes derived from our limited dataset may not be possible. The following edits to the text have been included to accommodate the reviewer’s concern:

LN 187: “Fits for the α parameter suggest a moderate to strong directional selection ($-\log \alpha \ll 0$) towards “attractor” phenotypic values (Table 1).”

LN 197: “Support for OU patterns is commonly interpreted as evidence for either stabilizing selection or phylogenetic niche conservatism (Cooper et al. 2016). However, perhaps a more parsimonious interpretation for our results would be directional selection—i.e. towards attractor phenotypes—having acted together with a physiological barrier, constraining the evolution of thermal tolerances beyond certain values. Our limited sample from the tree of life calls for caution trying to infer evolutionary processes from these results, which unequivocally inform of a strong phylogenetic structuring of both tolerance to heat and cold.”

Line 161 and 162. I think that this is an interpretation that does not belong in the results section. The models did not directly examine “adaptation to survive rare and extreme current climatic events” but instead looked at effects of clade age, paleo temperature, and current temperatures.

Re: We agree and to avoid over interpretation in the results section we have rewritten this sentence so that it now reads “adaptation to current climatic extremes”.

Line 201. Similar to my concerns described two comments previously. I think this statement

should be softened. For example, “attractor” could be a more appropriate term than “optimum”.

RE: We have replaced “optimum” with “attractor”.

Reviewers' Comments:

Reviewer #1:

Remarks to the Author:

I find this to be an interesting paper, which formalizes and generalizes some well known patterns, but adds new ones. The paper has clearly evolved in positive ways. The authors have done a good job of responding to suggestions and concerns.

I still have many relatively minor concerns (below). In some ways, I feel the inclusion of both ectotherms and endotherms to be comparing apples and oranges, as what sets limits on thermal neutral zones and on tolerance zones have nothing in common physiologically or evolutionarily. On the other hand, including both has some advantages

The ms. is largely pattern oriented, not mechanism oriented. However, patterns often promote subsequent mechanistic studies. Still I would have liked to have seen some speculation as to the mechanism underlying patterns or differences among groups. For example, the rapid evolution of "cold edges of the thermal neutral zone" of endotherms (relative to cold tolerance of ectotherms and plants) (see line 57 Abstract) probably relates to the ease of altering fur length or body size of endotherms. In contrast, evolutionary changes in cold tolerance of ectotherms might involve a series of interacting biochemical changes. Of course, I realize that Nature has word limits on mss, which constrains opportunities for discussion.

My main concern is with the conclusion that an "attractor" implies a "physiological evolutionary boundary." I may misinterpret "alpha" in the O-U analysis, but I don't think it measures a boundary in the definition of boundary. I try to explain this below in my comments (below).

Note, however, that after writing my review I took another look the comments of the 2nd reviewer and the responses. The reviewer did a much better job of explaining the concern. I disagree with the response of the authors (changing "strong physiological boundaries" to "physiological boundaries"). In my view, it is not a boundary. I'm unaware of any comparative analysis that determines whether an impenetrable or semi-impenetrable barrier exists. O-U does not.

I am confused by the way the data are presented. In Fig. 1, aquatic and terrestrial groups are separated, but not in subsequent figures and possibly analyses. Is there a reason for this? I don't recall discussion as to might cause a different pattern for aquatic vs. terrestrial species.

51-52 unclear"

56 "In the more recently originated endotherms... awkward phrasing and unclear. Do you mean that endotherms are more recently 'originated' than ectotherms, or do you meant, that among all endotherms, you are discussing only the more recently originated' ones? I assume the former.

59 logic unclear

74 performance/fitness curves use body temperature, not environmental temperature

81 add Wake et al., but the idea is old, tracing at least to Bogert (Evol. 1949) and Simpson. You cite Bogert later, and should probably cite it here, too.

96 you must cite W. J. Hamilton, Jr. — his maxithermy hypothesis was the first a physiological boundary hypothesis for heat tolerance. [I'm assuming here that you will choose to continue referring to physiological boundaries. If not, then you needn't cit Hamilon.]

fig. 1a on the pdf, distinguishing the age categories was tough, because of overlapping points.

Perhaps jitter or using a repel-program help here?

Fig. 1 the density plots to the right of the panels are hard to interpret (I cannot tell which density curve belongs to which group). Also, the legend needs to inficate that the horizontal lines are (presumably) the median value for a given group.

heterogeneously distributed in space or time?
not immediately obvious what "these" is referring to. In this paragraph, Introduce the limits furst,

then give distributions. You've done the reverse.

is a threshold a limit? or do you mean something different? Be consistent w/ terms.

152-3 interesting, but not obvious to my eye in Fig. 1b. In terrestrial ectotherms, there are very few recent origins, so one would expect less variation just by sampling.

add percentage (as you did for those originating in warm climates)

160-169 no mention of any statistics here. Throughout you make statement of differences but do not provide statistical backing. If these analyses are in the Supplement, then refer there. Otherwise, a reader will assume you do not have statistical backing.

Fig. 2 draw a vertical line, separating aquatic and terrestrial parts of each panel.

Fig. 2 perhaps draw dotted lines connecting, for example, CTmax of glaciated vs. warm aquatic ectotherms, and the same for the other comparisons. Easier to see the pattern.

*** Fig. 2 plants. You are getting upper thermal limits $> 50^{\circ}\text{C}$ — that is really hot. Are these from hot springs? I wonder if these samples are really representative of plants.

Also, you have a plant with a lower thermal boundary of almost -40°C . Is this a survival limit or a functional limit? Was this plant supercooled? My question here is are the end points in the various studies really comparable? Many terrestrial ectotherms can supercool, but they are not functional organisms at that point. I just am dubious that any "plant" is functional at -30°C , even if it can survive. My concern here is that your different lower limits may physiologically heterogeneous.

"beyond conservatism.. clade origin" — necessary phrase?
I'm not sure what you mean by "this appears not to apply" — better to say what applies. "This" could be referring to the preceding sentence, or also to the two preceding sentences.

Table 1. Why have plants and also plants & algae? why not simply plants and separately algae?

Table 1. Unites? Need to explain tempo, mode, etc. in the legend.

*** 171 does it necessarily imply a boundary, or a resistance? If a boundary, should should see a strongly left-skewed distribution, correct?

the wording is unclear — I think you are trying to write that lower limit of the TNZ is inversely related to the thickness of fur. but "documented an effect of" is a non-directional statement, so this is confusing.
readers unfamiliar with the concept of an attractor may wonder what you mean.
first mention of different tolerance metrics

*** 194-6 This is interesting, but is this a qualitative statement, or do you have statistical support? It reads like the former. I appreciate the need for conciseness in Nature papers, but (as I noted above) the text sometimes makes assertions with no statistical backing.

I don't follow your logic or why this is a "more parsimonious interpretation"

203-4 awkward sentence

awkward sentence

Fig 4 Is this for terrestrial organisms only, or did you pool terrestrial and aquatic. The most obvious pattern here is that palaeo-temperature seems relatively unimportant.

delete "or more important than" unless you have a statistical test
I'm having trouble following the logic here. fig 4 gives proportion of variance, not a coefficient of change. This may be in the Supplement.

Fig.4 Should you comment on the fact that R2 for endotherms are very low?

*** 219 You have 3 groups. Do you need to adjust significance levels for having done multiple comparisons?

"pace of evolution" means? use terminology used in the text
"at a MUCH lesser extent"
does it explain degree of "variability" or the value? Very different concepts
But don't most comparative studies deal with restricted clades, which will be on much shorter time scales. Is Brownian motion rejected in those studies?
"given the expected fitness benefits of surviving extreme conditions, — unclear, and I don't recall seeing any explicit analysis of extremes
"towards and upper physiological boundary." This is an important sentence but is confusing for several reasons. First, as I noted above, I do not understand why you see a boundary rather than something like an optimum in a stabilizing selection model. Perhaps the O-U methodology deals with this, but I'm dubious.

Let me give you an example. Lots of physiologists try to model the maximum performance of athletes (e.g, speed of 100-m dash). It is clear from the data and analyses that current records must be close to the boundary of what is humanly possible, assuming no changes in running surfaces, shoes, etc. In thermal biology, Hamilton argued that hotter was better, such that selection favors higher T_{opt} , but that animals could not evolve tolerance for very high temperatures. That is, there's a physiological limit or boundary.

To me an attractor is more like the optimum value, not a boundary.

If I'm misinterpreting what you mean by a boundary, then others might as well.

Incidentally, does the OH alpha parameter attract from both directions, or a single direction? If both higher and lower values converge, then then alpha is not measuring a boundary.

also, the wording here literally applies to both upper and lower thermal limits and for all groups (sentence starts with evolution of thermal limits," which implies both limits). I don't think you mean that.

Again if you are going to argue for a boundary (for upper limit), then you need to cite Hamilton. Granted he was largely using his intuition, but he was explicit about the idea of an upper thermal boundary (not an optimum) for animals.

cite Pötner?

ref 30. Stillman did not study tropical crabs in this paper, and he concluded that he species most at risk from warming were from the Northern Gulf of California, so this citation is incorrect.

Did the majority of species evolve in warm environments, or the majority of clades evolve there (as per Fig.1). Those aren't necessarily the same.
add some citations after "increases in thermal breadth"
interesting ideas, but greater selection opportunities or just stronger selection? Also, "reverse speciation gradients" is jargon needs clarification
Is climate linked with tolerance limits (as written) or the reverse?
This sentence seems inconsistent with the previous statement that tolerance limits are linked with climate.

292- unclear

unclear. What happens if there's only a lower boundary? Did you include this point in the analysis of lower tolerance limits?
"time for speciation effect"?
"homogeneous units of comparison" — do you mean homogeneous at the taxonomic level?

327 Have "critical thermal limits" been defined? Is this necessary?

A few papers report lethal and critical limits for single species (Brattstrom, Comp Biochem Physiol 1970) — lethal is higher than onset of spasms than is loss of righting response). thus lumping all will add noise, making it harder to detect patterns. So the fact that you see pattern when you lump probably implies that the patterns are robust.

Overall, I like the paper and think the paleo-perspective is a strong addition. My comments are intended mainly to make the paper more easily interpretable.

Ray Huey

Reviewer #2:

Remarks to the Author:

This is my third time seeing this MS, and I think that it is very much improved from the first version. All of my major critiques in both methodology and interpretation have been well addressed. I commend the authors on their thorough revisions throughout this long process.

I now only have a few minor comments that all pertain to readability. There are still a few places in the MS that I think are difficult to follow. I have attempted to outline them all in my line comments below. After these revisions for clarity, I think that this work will make a fine contribution to the field and be cited within the vast majority of introduction sections for thermal biology manuscripts for years to come.

Line Comments

Main MS

Ln 48: I find this first sentence confusing. In particular, the first clause, "Unveiling what determines the evolution of..." bit I find challenging to follow. Consider rewriting for clarity.

Lns 80-81: A noun appears to be missing after "ancestral climatic" rendering the sentence difficult to interpret. Please revise for clarity.

Lns 183-185: This appears to be the first mention of the evolutionary models test in the MS. As such, it somewhat comes out of the blue. I would prefer to have this developed in the introduction where you describe how you tested each of the hypotheses (possibly within the paragraph that begins with LN91).

LNS 203-204: I find the clause "which unequivocally inform of a strong phylogenetic structuring" difficult to follow. I think that I know what you are trying to state, but am not 100% certain. Please revise for clarity.

LNS 212-216: I found this sentence difficult to follow. I had to read it about 4 times to follow. Unfortunately, I am uncertain how best to fix. Initially, I misinterpreted the sentence as describing a comparison between ectotherms+endotherms and plants, when it is actually contrasting two potential drivers of thermal tolerance evolution only within ectotherms and endotherms. I describe all this just to give you a clue about one reader's confusion. Consider revising for improved clarity.

LN 262: I disagree that the findings are consistent with greater selection opportunities in cold environments. Warm environments should still exert strong selection (possibly even stronger) but there appears to be little additive genetic variance/heritability for such selection to result in evolutionary change. I suggest changing "greater selection opportunities" to "greater evolutionary potential"

LNS 284-286: This description of GlobTherm leaves out plants. Please add for clarity, or if plants aren't in GlobTherm but were in another database, please clarify.

Supplement

LNS102-103: I cannot follow this sentence as written. Most of my confusion comes from not knowing what "it" refers to, at least with any confidence. Please revise for clarity.

REVIEWER COMMENTS

Reviewer #1 (Remarks to the Author):

I find this to be an interesting paper, which formalizes and generalizes some well known patterns, but adds new ones. The paper has clearly evolved in positive ways. The authors have done a good job of responding to suggestions and concerns.

Response: We would like to thank reviewer #1 for the time spent on a comprehensive and thorough revision, which has helped us to improve our manuscript. Please see our detailed responses to all comments below.

I still have many relatively minor concerns (below). In some ways, I feel the inclusion of both ectotherms and endotherms to be comparing apples and oranges, as what sets limits on thermal neutral zones and on tolerance zones have nothing in common physiologically or evolutionarily. On the other hand, including both has some advantages

The ms. is largely pattern oriented, not mechanism oriented. However, patterns often promote subsequent mechanistic studies. Still I would have liked to have seen some speculation as to the mechanism underlying patterns or differences among groups. For example, the rapid evolution of “cold edges of the thermal neutral zone” of endotherms (relative to cold tolerance of ectotherms and plants) (see line 57 Abstract) probably relates to the ease of altering fur length or body size of endotherms. In contrast, evolutionary changes in cold tolerance of ectotherms might involve a series of interacting biochemical changes. Of course, I realize that Nature has word limits on mss, which constrains opportunities for discussion.

Response: We now discuss the valid point raised by the review in LN 307-310, which now reads “Evolutionary changes in cold tolerance of ectotherms might involve a series of interacting biochemical changes, which may take much longer to evolve than changes in fur length, feather depth or body size in endotherms.”

My main concern is with the conclusion that an “attractor” implies a “physiological evolutionary boundary.” I may misinterpret “alpha” in the O-U analysis, but I don’t think it measures a boundary in the definition of boundary. I try to explain this below in my comments (below).

Note, however, that after writing my review I took another look the comments of the 2nd reviewer and the responses. The reviewer did a much better job of explaining the concern. I disagree with the response of the authors (changing “strong physiological boundaries” to “physiological boundaries”). In my view, it is not a boundary. I’m unaware of any comparative analysis that determines whether an impenetrable or semi-impenetrable barrier exists. O-U does not.

Response: Yes, great point, since we can’t distinguish between an attractor and a boundary, we have acknowledged that our results support either scenario throughout the manuscript. Specifically, we have revised LN 104 and LN 269, which now read as below.

LN 104: “Specifically, if a boundary exists we would expect thermal physiological limits across clades to accumulate through time consistent with an Orstein-Uhlenbeck (OU) model of evolution which indicates a stabilizing selection on species thermal tolerance traits towards a fitness optimum²¹. Conversely, if neither a physiological boundary nor fitness optimum exist, the evolution of thermal limits may better fit a random model of evolution such as Brownian motion, where a trait evolves in a random walk process, or white noise, where the trait value varies independently around the global mean²².”

LN 269 “Instead, our results indicate evolution of upper thermal limits towards an attractor following an Orstein-Uhlenbeck model (Table 1), which is consistent with a scenario of selection

towards an upper physiological boundary that is not readily crossed, or an optimum beyond which fitness declines²⁰”

I am confused by the way the data are presented. In Fig. 1, aquatic and terrestrial groups are separated, but not in subsequent figures and possibly analyses. Is there a reason for this? I don't recall discussion as to might cause a different pattern for aquatic vs. terrestrial species.

Response: Figures 1 and 2 were designed to show the distribution of our data (we also note that the patterns in Figure 2 were obtained by analysing terrestrial and aquatic species separately). However, not describing how our data were distributed across aquatic and terrestrial groups in the text and subsequently pooled was an oversight, and we thank the reviewer for allowing us to rectify this. Revision made LN 161 “The distribution of data across aquatic and terrestrial realms reflects the distribution of life on earth where ~80 % of macroscopic species are terrestrial²⁵(Fig. 1b), because of the differences in data resolution across realms, and our search for general drivers, we pooled marine and terrestrial data together (except for plants) for subsequent analyses” Analyses were conducted on plants with and without aquatic plants throughout the manuscript (see table 1) and supplementary material.

51-52 unclear”

Response: We have revised LN 53 for clarity, which now reads “– acting as physiological evolutionary boundaries –“

56 “In the more recently originated endotherms... awkward phrasing and unclear. Do you mean that endotherms are more recently 'originated' than ectotherms, or do you meant, that among all endotherms, you are discussing only the more recently originated' ones? I assume the former.

Response: Great point. We have revised LN 58-61 for clarity, which now reads “In endotherms, the cold edges of the thermal neutral zone have evolved remarkably quickly compared to cold tolerance in ectotherms and plants, which may reflect the more recent origin of endotherms.”

59 logic unclear

Response: We have revised LN 61-64 for clarity “If the past tempo of evolution for upper thermal limits continues, adaptive responses in thermal limits will have limited potential to rescue the large majority of species under climate change”

74 performance/fitness curves use body temperature, not environmental temperature

Response: The suggested revision has been made. LN 78 now reads “body temperature increases at higher temperatures have a much greater effect on fitness than the equivalent temperature decreases at lower temperatures.”

81 add Wake et al., but the idea is old, tracing at least to Bogert (Evol. 1949) and Simpson. You cite Bogert later, and should probably cite it here, too.

Response: The suggested references have been added to LN 85

96 you must cite W. J. Hamilton, Jr. — his maxithermy hypothesis was the first a physiological boundary hypothesis for heat tolerance. [I'm assuming here that you will choose to continue referring to physiological boundaries. If not, then you needn't cite Hamilton.]

Response: The suggested citation has been added

fig. 1a on the pdf, distinguishing the age categories was tough, because of overlapping points. Perhaps jitter or using a repel-program help here?

Response: We have now improved the visibility of map by making the background darker.

Fig. 1 the density plots to the right of the panels are hard to interpret (I cannot tell which density curve belongs to which group). Also, the legend needs to indicate that the horizontal lines are (presumably) the median value for a given group.

Response: We have now improved the visibility of density plots by tweaking their transparency and added required details to the legend. We have also added a new Figure S1 to the supplementary file to more clearly show the relationships between order age in million years and lower and upper thermal tolerance limits.

heterogeneously distributed in space or time?

Response: We have clarified the text in LN 158-160, which now reads “Thermal physiological limits in our dataset are homogeneously represented across latitudes (ranging from 70° S to 70° N), while thermal ancestry is heterogeneously distributed across latitudes (Fig. 1a).”

148 not immediately obvious what “these” is referring to. In this paragraph, introduce the limits first, then give distributions. You've done the reverse.

Response: For clarity we have revised LN 168, which now reads “We compare evolutionary patterns in upper and lower thermal physiological limits for lethal and critical thermal limits of plants and ectotherms, and edges of thermal neutral zones separately as they each interact differently with species’ physiology, behaviour, and environmental conditions (see SI).”

150 is a threshold a limit? or do you mean something different? Be consistent w/ terms.

Response: For consistency we have changed “threshold” to “limit” see above revision.

152-3 interesting, but not obvious to my eye in Fig. 1b. In terrestrial ectotherms, there are very few recent origins, so one would expect less variation just by sampling.

Response: We have updated to the text to incorporate the reviewers suggestion, LN 173 now reads “However, this variation may be due to sampling as *n* decreases in both ectotherms and endotherms towards most recent times.”

156 add percentage (as you did for those originating in warm climates)

Response: We have added percentages to the text in LN 175, which now reads “Species with ancestors that originated under glaciation times are mostly sampled across temperate latitudes (~80 %) and in the Northern hemisphere (~80%) (Fig. 1).”

160-169 no mention of any statistics here. Throughout you make statement of differences but do not provide statistical backing. If these analyses are in the Supplement, then refer there. Otherwise, a reader will assume you do not have statistical backing.

Response: The reviewer is correct, we assumed the figure was simple enough to be self-explanatory, but statistical backing makes the results stronger. We have added a table to the supplement summarizing these results Supplementary table S1.

Fig. 2 draw a vertical line, separating aquatic and terrestrial parts of each panel.

Response: Done.

Fig. 2 perhaps draw dotted lines connecting, for example, CTmax of glaciated vs. warm aquatic ectotherms, and the same for the other comparisons. Easier to see the pattern.

Response: Done.

*** Fig. 2 plants. You are getting upper thermal limits > 50°C — that is really hot. Are these from hot springs? I wonder if these samples are really representative of plants.

Response: The majority of the upper temperature tolerance in plants >50 are for desert dwelling species. There were no hot spring plants sampled in the dataset.

Also, you have a plant with a lower thermal boundary of almost -40°C. Is this a survival limit or a functional limit? Was this plant supercooled? My question here is are the end points in the various studies really comparable? Many terrestrial ectotherms can supercool, but they are not functional organisms at that point. I just am dubious that any “plant” is functional at -30 °C, even if it can survive. My concern here is that your different lower limits may physiologically heterogeneous.

Response: The low values for terrestrial plants are species of moss, lichen and liverwort. The estimates of thermal tolerance approaching -40°C for some of these species was estimated using the most common measure of lower thermal tolerance is terrestrial plants, which is the temperature at which 50% of plant material dies, most commonly measured as the point when 50% of photosynthetic capacity is lost when temperatures are returned to ambient conditions (i.e. tissue survival of the extreme cold). We agree that comparability across limits is a pervasive issue in macrophysiology, which is why we included specific sensitivity analyses to test for effects derived from non-homologous thermal limit types (i.e. survival vs. critical limits, see SI Tables S3, S5), and have analysed plants separately given the tendency for plant eco-physiologists to focus on tissue rather than whole-organisms survival. Our results are meant to focus on comparisons within similar metrics and groups rather than across those metrics, and like all synthesis studies, rely on the assumption that further study-level variation is homogeneous rather than biased across our studied variables. We have clarified the importance of this assumption in the discussion of caveats and limitations section of the supplementary material LN 254 “Different experimental measures of thermal tolerance limits are associated with different taxonomic groups. These differences between experimental estimates of thermal tolerance is why we focused our comparisons on similar metrics and groups rather than across those metrics. Endotherms are all measured using TNZ, plants and algae are almost exclusively lethal experiments, and in ectotherms are majority of experiments measure critical limits. Plants were also analysed separately because, experiments with plants tend to focus on tissue rather than whole-organisms survival. Like all synthesis studies, our study relies on the assumption that further study-level variation is homogeneous rather than biased across our studied variables.”

“beyond conservatism.. clade origin” — necessary phrase?

Response: LN 187 has been revised for clarity and now reads “factors other than conservatism of climatic conditions must shape the variation in these thermal traits.”

I’m not sure what you mean by “this appears not to apply” — better to say what applies. “This” could be referring to the preceding sentence, or also to the two preceding sentences.

Response: We have clarified the sentence in LN 189, which now reads: “However, a colder origin of ancestry appears not to apply, broadly speaking, to endotherms and aquatic plants (Fig. 1b, Supplementary table S1.”

Table 1. Why have plants and also plants & algae? why not simply plants and separately algae?

Response: The rationale was due to taxonomic resolution and analysis power, which was rather limited in the case of algae (and even when both groups are analysed together). The reason to analyse plants separate from algae was due to these taxa being classified in different taxonomic kingdoms and thus the need to check for the effects of mixing pears and apples.

Table 1. Unites? Need to explain tempo, mode, etc. in the legend.

Response: We thank the reviewer for spotting this. The legend now reads: “Tempo and mode of evolution of upper and lower thermal tolerances of ectothermic species, endothermic species, terrestrial plants and plants and algae. The tempo measures the rate of thermal tolerance

evolution (in °C/Mya), and the mode informs of the likelihood within which a given model of evolution fits the data (for details, see Section 2 in Supporting Information).

*** 171 does it necessarily imply a boundary, or a resistance? If a boundary, should should see a strongly left-skewed distribution, correct?

Response: We have changed to “physiological constraints” here.

the wording is unclear — I think you are trying to write that lower limit of the TNZ is inversely related to the thickness of fur. but “documented an effect of” is a non-directional statement, so this is confusing.

Response: LN 302 has been revised for clarity and now reads “biophysical models have shown that the lower limits of TNZ in mammals decrease with increasing body size and the thickness of fur insulation³⁶, but no relationship of size on CTmin exists in lizards³⁷.”

readers unfamiliar with the concept of an attractor may wonder what you mean.

Response: We have now added a clarification in LN 211 that reads:

“-i.e. or phenotypic values with non-random higher frequencies, suggesting they are selected for”

first mention of different tolerance metrics

Response: The thermal tolerance metric used for each taxonomic group are first mentioned in the first paragraph of the results. We have updated the text in LN 165 to improve clarity and it now reads “Thermal physiological limits include lethal and critical thermal limits of plants and ectotherms, and edges of thermal neutral zones in endotherms (for more details see methods and supplementary section 1).”

*** 194-6 This is interesting, but is this a qualitative statement, or do you have statistical support? It reads like the former. I appreciate the need for conciseness in Nature papers, but (as I noted above) the text sometimes makes assertions with no statistical backing.

Response: The statistical backing is provided by the parameter estimations in Table 1 and the Tables referenced in Section 5 of the SI. Further, more comprehensive statistical backing is provided by Section 2.3 of the SI. For clarity we have updated the text in LN 218 “The patterns of mode of evolution show that selection towards an ‘attractor’ is stronger on upper thermal limits than lower thermal limits in endotherms and terrestrial plants (lower $-\log \alpha$ in Table 1), but a comprehensive comparison is limited by data availability (see Section 2.3 and 5 in Supporting Information).”

I don’t follow your logic or why this is a “more parsimonious interpretation”

Response: The rationale is that interpretations based on either stabilizing selection or PNC would involve additional mechanisms (for which our data does not allow testing). Instead, the higher likelihood of OU models in our results could be simply explained by upper thermal limits being more physiologically constrained to evolve far beyond given phenotypic values.

203-4 awkward sentence

Response: We have revised LN 226 for clarity, which now reads “Our results clearly show strong phylogenetic structure in both tolerance to heat and cold, however, our data is only a limited sample of the full tree of life and thus, we recommend caution when trying to infer evolutionary processes from our results.”

awkward sentence

Response: We have revised LN 234 for clarify, “Using random forests to compare all three hypotheses invoked to explain thermal limits we found current minimum and maximum

environmental temperatures experienced by species²⁸, to play a strong role in determining thermal tolerance variation, consistent with the importance of *adaptation to current climatic extremes* (Fig. 4).”

Fig 4 Is this for terrestrial organisms only, or did you pool terrestrial and aquatic. The most obvious pattern here is that palaeo-temperature seems relatively unimportant.

Response: We have updated the caption for Figure 4 to state that we combined data from aquatic and terrestrial realms. And yes, your interpretation of the figure is correct in that paleoclimate was the least important variable as stated in LN 248 “Palaeoclimate ranked the lowest across all factors tested and only emerged as a significant variable for ectotherms: for which, however, its importance reached 5.9-5.3% for upper and lower limits, respectively (Fig. 4a, Supplementary Figure S5).”

delete “or more important than” unless you have a statistical test

Response: The statistical results are in Table S4 and S5, the text has been updated to indicate this.

I’m having trouble following the logic here. fig 4 gives proportion of variance, not a coefficient of change. This may be in the Supplement.

Response: Thank you for giving us a chance to rectify this error, the statistical results which support this text are in Table S4 and S5. The text throughout this section has been updated to indicate this.

Fig.4 Should you comment on the fact that R2 for endotherms are very low?

Response: Our guess is that lower R2 for endotherms has to do with their weaker gradients in thermal tolerance and perhaps with the multiple mechanisms (e.g. behaviour shifts, fat accumulation, body size clines) that would make them less dependent on ambient temperature or would have allowed overcoming palaeo-origin constraints. This explanation is purely speculative and thus we had not included it in the main text. We have updated the results in LN240 “the proportion of variance explained in endotherms was much lower than for ectotherms and plants (see Supplementary Figure S5”

*** 219 You have 3 groups. Do you need to adjust significance levels for having done multiple comparisons?

Response: In the random forest analyses referred to here (Fig. 4a), we did not conduct frequentist significance comparisons, and thus, in principle such analyses would not be subjected to adjusting/correcting p-values. If requested, we could conduct means comparisons across predictors and upper/lower limits, applying a Bonferroni correction for multiple tests. However, we have not deemed it necessary because the aim of the analyses is to rank predictors and explore patterns according their accuracy and, in our opinion, results are sufficiently clear to that end.

To clarify, when we referred to significance in line 220, it was relative to the importance of a randomly generated predictor (across, e.g. 100 iterations). This procedure, of common use in the random forests literature, is useful to determine whether or not the level of importance of any of the predictors in our models could be reached by chance alone. Please see below a version of Fig. 4 including the ‘null’ level of importance reached by randomly generated predictors (at their upper 95% CI) represented by dashed lines (blue for lower limits, red for upper limits). We would be willing to replace the original Fig. 4 by this version if the editor deems it convenient.

“pace of evolution” means? use terminology used in the text

Response: pace of evolution has been used in the literature analogous to tempo of evolution, implying the rate or speed at which a given phenotype evolves. We however, realize it can be confusing and thus, have replaced “pace” for “tempo”.

“at a MUCH lesser extent”

Response: suggested revision was made LN 260

does it explain degree of “variability” or the value? Very different concepts

Response: We have revised LN 258, which now reads “Together, our results show that present-day environmental temperatures, but also that physiological constraints in the pace of evolution (in ectotherms and endotherms) and, to a much lesser extent, the climate at clade origin (for cold limits in ectotherms) affect cold and heat tolerances and thus could set limits on species distributions.”

But don't most comparative studies deal with restricted clades, which will be on much shorter time scales. Is Brownian motion rejected in those studies?

Response: This is a really interesting question but answering it would require us to conduct a meta-analysis, more so given the vast literature on comparative studies. Without doing so, we would not want to say in which proportion BM models are rejected (nor whether cross-model comparisons are done appropriately).

"given the expected fitness benefits of surviving extreme conditions, — unclear, and I don't recall seeing any explicit analysis of extremes

Response: We have updated the text to use a more consistent definition with our "adaptation to current climatic extremes hypothesis", LN 270 now reads "which is consistent with a scenario of selection towards an upper physiological boundary that is not readily crossed, or an optimum beyond which fitness declines²⁰."

"towards and upper physiological boundary." This is an important sentence but is confusing for several reasons. First, as I noted above, I do not understand why you see a boundary rather than something like an optimum in a stabilizing selection model. Perhaps the O-U methodology deals with this, but I'm dubious.

Let me give you an example. Lots of physiologists try to model the maximum performance of athletes (e.g., speed of 100-m dash). It is clear from the data and analyses that current records must be close to the boundary of what is humanly possible, assuming no changes in running surfaces, shoes, etc. In thermal biology, Hamilton argued that hotter was better, such that selection favors higher T_{opt} , but that animals could not evolve tolerance for very high temperatures. That is, there's a physiological limit or boundary.

To me an attractor is more like the optimum value, not a boundary.

If I'm misinterpreting what you mean by a boundary, then others might as well. Incidentally, does the OH alpha parameter attract from both directions, or a single direction? If both higher and lower values converge, then then alpha is not measuring a boundary.

Response: As the reviewer notes, both higher and lower values converge towards an "attractor", but more narrowly so for upper than lower thermal limits. This is, upper limits are not as 'free' to evolve variation, which we interpret this as a boundary. The reviewer is correct that we can't distinguish between an attractor and a boundary, we now acknowledge that our results support either scenario.

LN 269: "Instead, our results indicate evolution of upper thermal limits towards an attractor following an Orstein-Uhlenbeck model (Table 1), which is consistent with a scenario of selection towards an upper physiological boundary that is not readily crossed, or an optimum beyond which fitness declines²⁰,"

also, the wording here literally applies to both upper and lower thermal limits and for all groups (sentence starts with evolution of thermal limits," which implies both limits). I don't think you mean that.

Response: We have revised LN 269 for clarity, which now reads "Instead, our results indicate evolution of upper thermal limits."

Again if you are going to argue for a boundary (for upper limit), then you need to cite Hamilton. Granted he was largely using his intuition, but he was explicit about the idea of an upper thermal boundary (not an optimum) for animals.

Response: The suggested revision was made and Hamilton 1973 has been added to LN 315.

cite Pörtner?

Response: The suggested revision was made and Pörtner 2001 has been added to LN 278.

ref 30. Stillman did not study tropical crabs in this paper, and he concluded that the species most at risk from warming were from the Northern Gulf of California, so this citation is incorrect.

Response: The suggested revision was made and the Stillman citation has been removed.

Did the majority of species evolve in warm environments, or the majority of clades evolve there (as per Fig.1). Those aren't necessarily the same.

Response: Thank you for drawing this to our attention, LN 288 has been revised and now reads, "the majority of clades evolved in warm environments."

add some citations after "increases in thermal breadth"

Response: We now reference supplementary table S3 which provides statistical support for this claim.

interesting ideas, but greater selection opportunities or just stronger selection? Also, "reverse speciation gradients" is jargon needs clarification

Response: Suggested revisions have been made in LN 294, which now reads "These findings would be consistent with the existence of stronger greater evolutionary potential in cold environments and perhaps with the reverse speciation gradients i.e. lower species diversity in the tropics35."

Is climate linked with tolerance limits (as written) or the reverse?

Response: Suggested revision made LN 318, "Species thermal tolerance limits appears to be strongly linked to current climate".

This sentence seems inconsistent with the previous statement that tolerance limits are linked with climate.

Response: We have revised LN316 for clarity, which now reads "We find that these three mechanisms all effect species thermal tolerance limits , but their relative importance varies. Specifically, species thermal tolerance limits appears to be strongly linked to current climate, but there is also evidence supporting the existence of physiological boundaries to the evolution of upper temperature tolerance across all groups, and a small (but consistent) effect of temperature of clade origin in cold tolerance for ectotherms."

292- unclear

Response: We have clarified LN 345, which now reads: "GlobTherm only contains lethal limit data from studies where all temperature treatments were indicated, and only contains thermal neutral zone data from studies showing evidence that the upper or lower boundary of the thermal neutral zone was reached."

unclear. What happens if there's only a lower boundary? Did you include this point in the analysis of lower tolerance limits?

Response: see our response to previous comments. The dataset contains species for which only one limit (either upper or lower) had been recorded. We included those species in the analyses to avoid loss of analytical power.

"time for speciation effect"?

Response: To clarify our meaning, we have revised LN370, which now reads:

“To better disentangle the effects of temperature at clade origin from a time for speciation effect—i.e. the fact that clades that have existed for longer would be more diverse simply due to having had longer times to diversify (Stephens and Wiens 2003)—, results...”

“homogeneous units of comparison” — do you mean homogeneous at the taxonomic level?

Response: Yes, we have clarified the text in LN 378, which now reads “provide homogeneous units of comparison at the taxonomic level for phenotypic divergence.”

Have “critical thermal limits” been defined? Is this necessary?

Response: We now introduce the different metrics used much sooner in LN 337, which now reads “Lethal thermal limits mark the temperature when mortality occurs. Critical thermal limits record the temperature at which a key an ecological function is lost, such as locomotion, or as in the case of endotherms the ability maintain basal metabolism (i.e. TNZ).

A few papers report lethal and critical limits for single species (Brattstrom, Comp Biochem Physiol 1970) — lethal is higher than onset of spasms than is loss of righting response). thus lumping all will add noise, making it harder to detect patterns. So the fact that you see pattern when you lump probably implies that the patterns are robust.

Response: We completely agree with the reviewer

Overall, I like the paper and think the paleo-perspective is a strong addition. My comments are intended mainly to make the paper more easily interpretable.

Ray Huey

Reviewer #2 (Remarks to the Author):

This is my third time seeing this MS, and I think that it is very much improved from the first version. All of my major critiques in both methodology and interpretation have been well addressed. I commend the authors on their thorough revisions throughout this long process.

I now only have a few minor comments that all pertain to readability. There are still a few places in the MS that I think are difficult to follow. I have attempted to outline them all in my line comments below. After these revisions for clarity, I think that this work will make a fine contribution to the field and be cited within the vast majority of introduction sections for thermal biology manuscripts for years to come.

Line Comments

Main MS

Ln 48: I find this first sentence confusing. In particular, the first clause, “Unveiling what determines the evolution of...” bit I find challenging to follow. Consider rewriting for clarity.

Response: We have revised LN 49-51 for clarity, which now reads “Understanding what mechanisms determine how species thermal limits have evolved across the tree of life is central to predict species’ responses to climate change.”

Lns 80-81: A noun appears to be missing after “ancestral climatic” rendering the sentence difficult to

interpret. Please revise for clarity.

Response: We have revised LN 84-85 for clarity, which now reads “First, deep-time climate legacies would be detectable in thermal limits if species had a tendency to retain their ancestral climatic affinities under niche conservatism.”

Lns 183-185: This appears to be the first mention of the evolutionary models test in the MS. As such, it somewhat comes out of the blue. I would prefer to have this developed in the introduction where you describe how you tested each of the hypotheses (possibly within the paragraph that begins with LN91).

Response: We have updated our explanation of the *physiological boundaries* hypothesis to include the phylogenetic models used to test it, LN 104-111 now reads “Specifically, if a boundary exists we would expect thermal physiological limits across clades to accumulate through time consistent with an Orstein-Uhlenbeck (OU) model of evolution which indicates a stabilizing selection on species thermal tolerance traits towards a fitness optimum²¹. Conversely, if a physiological boundary does not exist, the evolution of thermal limits may better fit a random model of evolution such as Brownian motion, where a trait evolves in a random walk process or white noise, where the trait value varies independently around the global mean²².”

LNS 203-204: I find the clause “which unequivocally inform of a strong phylogenetic structuring” difficult to follow. I think that I know what you are trying to state, but am not 100% certain. Please revise for clarity.

Response: We have revised LN 226 for clarity, which now reads “Our results clearly show strong phylogenetic structure in both tolerance to heat and cold”

LNS 212-216: I found this sentence difficult to follow. I had to read it about 4 times to follow. Unfortunately, I am uncertain how best to fix. Initially, I misinterpreted the sentence as describing a comparison between ectotherms+endotherms and plants, when it is actually contrasting two potential drivers of thermal tolerance evolution only within ectotherms and endotherms. I describe all this just to give you a clue about one reader’s confusion. Consider revising for improved clarity.

Response: We have revised LN 241 for clarity, which now reads “However, for ectotherms and endotherms the age of the origin clade was similarly or more important than current climate conditions in explaining thermal tolerance limits (Fig. 4, Supplementary Table S4 and S5). This result is consistent with the role of constraints on the pace of evolution (part of the *physiological boundaries* hypothesis), because older clades have had more time to evolve.”

LN 262: I disagree that the findings are consistent with greater selection opportunities in cold environments. Warm environments should still exert strong selection (possibly even stronger) but there appears to be little additive genetic variance/heritability for such selection to result in evolutionary change. I suggest changing “greater selection opportunities” to “greater evolutionary potential”

Response: We have made the suggested change in LN 294, which now reads “These findings would be consistent with the existence of greater evolutionary potential in cold environments and perhaps with reverse speciation gradients i.e. lower species diversity in the tropics³⁵”

LNS 284-286: This description of GlobTherm leaves out plants. Please add for clarity, or if plants aren’t in GlobTherm but were in another database, please clarify.

Response: GlobTherm also contains data for plants, we have revised LN 333 for clarity “Experimentally-derived thermal tolerance limit data were obtained from GlobTherm²³, which assembles published measurements of upper and lower thermal tolerance limits, including both lethal and critical thermal metrics for plants and ectotherms and the edges of thermal neutral zones (TNZ) in endotherm.”

Supplement

LNS102-103: I cannot follow this sentence as written. Most of my confusion comes from not knowing what "it" refers to, at least with any confidence. Please revise for clarity.

Response: We have revised Supplementary material LN 102 -103, which now reads "We choose the TimeTree¹⁵ for our phylogenetic hypothesis because it is the largest, multi-taxon, more comprehensive dated phylogenetic tree to date."

Reviewers' Comments:

Reviewer #1:

Remarks to the Author:

I think the ms. is interesting, provocative, and much improved. The authors have done a good job of responding to almost all of my suggestions, though I still have concerns about boundaries vs. OU (see below). [Perhaps I'm missing something here.] Overall, this paper should be of interest to a broad audience.

I do think the authors need to make a quick pass through the text to clean up various issues (listed below). In particular, I encourage them to edit the Abstract.

Because of pressing deadlines, I did not re-review the Methods or Supplement.

Ray Huey

The abstract needs work

49 implies this paper will deal with mechanisms, but it actually deals almost exclusively with pattern.

51 type of data isn't specified. Mechanisms? Thermal limits?

52/4 seem to contradict 60/2. If thermal limits are adapted to current climate extremes, why wouldn't they track contemporary climate change?

59-60 The logic will be unclear, and I still hold that this is because fur length/thickness should evolve quickly (see my previous comments). You have adjusted the main text to cover this point, but but the Abstract still sticks with the idea that only time of origin is driving the pattern. Perhaps.

74 change taxonomy to phylogeny.

102 I may still be wrong, but I see a boundary limitation as philosophically different from stabilizing selection, as modeled by OU. Check Wikipedia for a simple description of OU, which is not a boundary process (to me at least). Perhaps current methods can't distinguish between selection pushing up against an impenetrable barrier versus selection towards and then stabilizing at an optimum.

110 slower than for cold tolerance?

159... edit

162 clump > lump, delete together (redundant)

169 variation within lower and within upper, or variation between upper and lower? In any case hard to see the pattern in Fig. 1b. Just a sample size issue, as per line 171? Isn't that testable?

172 "of most analyzed species" is confusing. To me, could imply that ancestors of most unanalyzed species originated under cold conditions, which is not what you mean. delete "analyzed"?

178 compared with

216/19 Please provide a justification for this statement. Also, I don't understand "beyond given phenotypic values"

"the age of the origin clade was similarly or more important than current ^{SEP} climate conditions in explaining thermal tolerance limits". But what is the direction of the effect? Not evident from Fig. 4 or

from the text. Are you saying that the older the clade, the more current climate explains thermal limits? or less? ^L_{SEP}
235/7 I'm lost.

big jump from tolerances to distributional limits. Provide a citation or two or delete the comment.
move "(Table 1)" so it follows lower thermal limits, because Table 1 does not address other comparative studies.

259 'indicate evolution of upper thermal limits towards'
indicate that upper thermal limits evolve towards
syntax — an attractor isn't following a OU model (wordy sentence, also)

inconsistent capitalization of titles in references (e.g., Sinervo et al., Caspermeyer et al.

in warm environments or during warm periods? those are different.
283/5 unclear

add period after endotherms.
seems like an overstatement here — in lizards, at least, several prior studies have looked at this, sometimes with a phylogenetic perspective. Do you mean broad phylogenetic scale here?
Huey et al. 2012 is more appropriate than Tewksbury.
odd font substitution in several successive titles (at least in the pdf)

Reviewer #1 (Remarks to the Author):

I think the ms. is interesting, provocative, and much improved. The authors have done a good job of responding to almost all of my suggestions, though I still have concerns about boundaries vs. OU (see below). [Perhaps I'm missing something here.] Overall, this paper should be of interest to a broad audience.

I do think the authors need to make a quick pass through the text to clean up various issues (listed below). In particular, I encourage them to edit the Abstract.

Because of pressing deadlines, I did not re-review the Methods or Supplement.
Ray Huey

The abstract needs work

49 implies this paper will deal with mechanisms, but it actually deals almost exclusively with pattern.

Revision: The reference to mechanisms has been removed from the abstract. LN 47 now reads “Understanding how species’ thermal limits have evolved across the tree of life is central to predicting species’ responses to climate change.”

51 type of data isn’t specified. Mechanisms? Thermal limits?

Revision: We now state more clearly the type of data we are using. LN 49 now reads “Here, using experimentally-derived estimates of physiological thermal limits in over 2000 terrestrial and aquatic species...”

52/4 seem to contradict 60/2. If thermal limits are adapted to current climate extremes, why wouldn’t they track contemporary climate change?

Revision: We have revised LN 61 for clarity, which now reads “If the past tempo of evolution for upper thermal limits continues, adaptive responses in thermal limits will have limited potential to rescue the large majority of species given the unprecedented rate of contemporary climate change.”

59-60 The logic will be unclear, and I still hold that this is because fur length/thickness should evolve quickly (see my previous comments). You have adjusted the main text to cover this point, but the Abstract still sticks with the idea that only time of origin is driving the pattern. Perhaps.

Revision: We have revised the abstract and have now removed the statement referring to time of origin as the driver of this pattern.

74 change taxonomy to phylogeny.

Revision made LN 76.

102 I may still be wrong, but I see a boundary limitation as philosophically different from stabilizing selection, as modeled by OU. Check Wikipedia for a simple description of OU, which is not a boundary process (to me at least). Perhaps current methods can't distinguish between selection pushing up against an impenetrable barrier versus selection towards and then stabilizing at an optimum.

Indeed, it is our understanding that current methods cannot distinguish between selection pushing up against a barrier versus selection towards and stabilizing at an optimum. i.e. Phenotypes beyond a barrier may not exist because they are not physiologically possible or because they are selected against, and the comparisons of the models cannot distinguish among these two possibilities.

110 slower than for cold tolerance?

Revision made: We have revised LN 111 for clarity, which now reads "While slower rates of evolution are expected for heat tolerance, because it is less variable than cold tolerance..."

159... edit

Revision made: We have revised LN 162 for clarity, which now reads "The distribution of data across aquatic and terrestrial realms reflects the distribution of life on earth where ~80 % of macroscopic species are terrestrial (Fig. 1b). Because of the differences in data resolution across realms, with less samples for aquatic taxa, we clump marine and terrestrial data (except for plants) for subsequent analyses."

162 clump > lump, delete together (redundant)

Revision made: Suggested revision made and we have removed "together" from LN 165. We have replaced the word "clump" with "pool" in LN 166.

169 variation within lower and within upper, or variation between upper and lower? In any case hard to see the pattern in Fig. 1b. Just a sample size issue, as per line 171? Isn't that testable?

Response: Yes, there is greater variability within lower and upper thermal tolerance in older clades. We have revised LN 174 to clarify this "We found that variation within lower and upper thermal limits increases with clade age, more clearly in ectotherms and endotherms than in plants (Fig. 1b)."

172 "of most analyzed species" is confusing. To me, could imply that ancestors of most unanalyzed species originated under cold conditions, which is not what you mean. delete "analyzed"?

Response: We have clarified the LN 177: "Most ancestors of the species in our dataset (~80%) originated under warm climatic regimes".

178 compared with

Response: done

216/19 Please provide a justification for this statement. Also, I don't understand "beyond given phenotypic values"

Good point. We have re-evaluated this argument and we no longer feel strongly about the parsimony argument. We have changed our wording to "alternative" in LN223.

233 "the age of the origin clade was similarly or more important than current climate conditions in explaining thermal tolerance limits". But what is the direction of the effect? Not evident from Fig. 4 or from the text. Are you saying that the older the clade, the more current climate explains thermal limits? or less?

Response: We thank the reviewer for spotting this and realize this was not sufficiently clear. The direction of the mentioned effect is documented in Supp. Fig. 6a-d, which shows that species belonging to older clades tend to tolerate more heat (in ectotherms and endotherms), and more cold in ectotherms. We now point the reader towards this supplementary result in the main text.

235/7 I'm lost.

Response: We realize the noted sentence was confusing and have deleted it.

252 big jump from tolerances to distributional limits. Provide a citation or two or delete the comment.

Revision made: We have deleted the comment in LN 262.

move "(Table 1)" so it follows lower thermal limits, because Table 1 does not address other comparative studies.

Suggested revision made.

259 'indicate evolution of upper thermal limits towards'

syntax — an attractor isn't following an OU model (wordy sentence, also)

Revision made: We have revised LN 269 for clarity and we have broken this sentence into two parts, which now read "Instead, our results indicate that upper thermal limits evolve towards an attractor value consistent with an Orstein-Uhlenbeck model of evolution (Table 1). This result is consistent with a hypothesis of selection towards an upper physiological boundary that is not readily crossed, or an optimum beyond which fitness declines."

inconsistent capitalization of titles in references (e.g., Sinervo et al., Caspermeyer et al.

The suggested revision has been made.

in warm environments or during warm periods? those are different.

Response: We have clarified the text in LN 288, which now reads "Because the majority of clades evolved during warm periods, the species-poor and cooler higher latitudes appear to offer opportunities for speciation and evolution of thermal tolerance through adaptive radiation".

283/5 unclear

We have revised LN 294 for clarity, which now reads "These findings are consistent with the hypothesis that there are greater opportunities for speciation and evolution of thermal tolerances in cold environments and perhaps with reverse speciation gradients i.e. lower speciation in the tropics³⁶."

add period after endotherms.

Suggested revision made.

seems like an overstatement here — in lizards, at least, several prior studies have looked at this, sometimes with a phylogenetic perspective. Do you mean broad phylogenetic scale here?

Revision made: Thank you for bringing this to our attention. Yes, we mean broad-scale study. We have revised LN 311 accordingly “Our study provides a broad-scale formal test of the longstanding hypotheses that species thermal limits are conserved..”

Huey et al. 2012 is more appropriate than Tewksbury.

The suggested revision has been made and Huey et al. 2012 has replaced Tewksbury 2002.

253 odd font substitution in several successive titles (at least in the pdf)

Corrected: the same font is now used throughout the text.